EMBO
Molecular Medicine

# Overcoming resistance to anabolic SARM therapy in experimental cancer cachexia with an HDAC inhibitor

Sophia G Liva[1,†], Yu-Chou Tseng[2,†], Anees M Dauki[1], Michael G Sovic[1], Trang Vu[1], Sally E Henderson[3], Yi-Chiu Kuo[2], Jason A Benedict[4], Xiaoli Zhang[4], Bryan C Remaily[1], Samuel K Kulp[1], Moray Campbell[1,5], Tanios Bekaii-Saab[6], Mitchell A Phelps[1,5], Ching-Shih Chen[2,7] & Christopher C Coss[1,5,*] (iD)

## Abstract

No approved therapy exists for cancer-associated cachexia. The colon-26 mouse model of cancer cachexia mimics recent late-stage clinical failures of anabolic anti-cachexia therapy and was unresponsive to anabolic doses of diverse androgens, including the selective androgen receptor modulator (SARM) GTx-024. The histone deacetylase inhibitor (HDACi) AR-42 exhibited anti-cachectic activity in this model. We explored combined SARM/AR-42 therapy as an improved anti-cachectic treatment paradigm. A reduced dose of AR-42 provided limited anti-cachectic benefits, but, in combination with GTx-024, significantly improved body weight, hindlimb muscle mass, and grip strength versus controls. AR-42 suppressed the IL-6/GP130/STAT3 signaling axis in muscle without impacting circulating cytokines. GTx-024-mediated β-catenin target gene regulation was apparent in cachectic mice only when combined with AR-42. Our data suggest cachectic signaling in this model involves catabolic signaling insensitive to anabolic GTx-024 therapy and a blockade of GTx-024-mediated anabolic signaling. AR-42 mitigates catabolic gene activation and restores anabolic responsiveness to GTx-024. Combining GTx-024, a clinically established anabolic therapy, with AR-42, a clinically evaluated HDACi, represents a promising approach to improve anabolic response in cachectic patients.

**Keywords** androgen; cachexia; HDAC inhibitor; selective androgen receptor modulator; STAT3
**Subject Categories** Cancer; Metabolism; Musculoskeletal System

## Introduction

Cancer cachexia is a multifactorial syndrome characterized by the involuntary loss of muscle mass occurring with or without concurrent losses in adipose tissue. The progressive loss of lean mass associated with cachexia results in decreased quality of life, decreased tolerance of chemotherapy, and reduced overall survival (Baracos et al, 2018). It is estimated that 50–80% of all cancer patients experience cachexia symptoms and up to 20% of all cancer-related deaths are attributable to complications arising from cachexia-mediated functional decline (Tisdale, 2009). A multitude of tumor and host factors are recognized as contributors to the multi-organ system dysfunction in cancer cachexia, presenting a considerable therapeutic challenge. Diverse cachexia treatment strategies have been evaluated in patients with few offering effective palliation and none gaining FDA approval for this devastating consequence of advanced malignancy (von Haehling & Anker, 2015). Among the complex sequelae associated with cachectic progression, compromised muscle function associated with reduced muscle mass is viewed as a primary contributor to patient morbidity and mortality (Baracos et al, 2018). Recognizing this feature of cancer cachexia, regulatory agencies require the demonstration of meaningful improvements in physical function in addition to improvements in patient body composition for successful registration of novel cachexia therapies (Fearon et al, 2015). Anabolic androgenic steroids or steroidal androgens are among the most well-recognized function-promoting therapies (Bhasin et al, 1996) and, as such, have been extensively evaluated in muscle wasting of diverse etiology (Orr & Fiatarone Singh, 2004). Despite meeting FDA approval criteria in other wasting diseases (Orr & Fiatarone Singh, 2004), steroidal androgens are yet to demonstrate clinical benefit in cancer cachexia. However, the continued development of novel androgens

1 Division of Pharmaceutics and Pharmacology, College of Pharmacy, The Ohio State University, Columbus, OH, USA
2 Division of Medicinal Chemistry and Pharmacognosy, College of Pharmacy, The Ohio State University, Columbus, OH, USA
3 Department of Veterinary Biosciences, College of Veterinary Medicine, Ohio State University, Columbus, OH, USA
4 Center for Biostatistics, Department of Biomedical Informatics, The Ohio State University, Columbus, OH, USA
5 The Ohio State University Comprehensive Cancer Center, The Ohio State University, Columbus, OH, USA
6 Mayo Clinic Cancer Center, Phoenix, AZ, USA
7 Department of Medical Research, China Medical University Hospital, China Medical University, Taichung, Taiwan
 *Corresponding author. Tel.: +1 614 688 1309; Fax: +1 614 292 7766; E-mail:coss.16@osu.edu
 †These authors contributed equally to this work

for the treatment of wasting diseases suggests confidence in this therapeutic strategy remains (von Haehling & Anker, 2015).

In addition to their well-characterized anabolic effects on skeletal muscle, steroidal androgens elicit a number of undesirable virilizing side effects and can promote prostatic hypertrophy, which limits their widespread clinical use (Coss *et al*, 2014). Recently developed, non-steroidal, selective androgen receptor modulators (SARMs) offer a number of improvements over steroidal androgens including prolonged plasma exposures and oral bioavailability with greatly reduced side effects (virilization, etc.), while maintaining full agonism in anabolic tissues such as skeletal muscle (Mohler *et al*, 2009). With once-daily dosing, the SARM GTx-024 (enobosarm) showed promising gains in fat-free mass in both male and female cancer patients, but ultimately failed to demonstrate a clear functional benefit in pivotal phase III trials in a cachectic non-small-cell lung cancer (NSCLC) population (Srinath & Dobs, 2014). GTx-024 has a strong safety profile and proven effects on skeletal muscle, but is no longer being developed for cancer cachexia.

Hypogonadism is a feature of advanced malignancy and experimental cachexia that worsens multiple cachectic sequelae, including decreased skeletal muscle mass, providing a rationale for therapeutic exogenous androgen administration (Vigano *et al*, 2010; White *et al*, 2013b). Though the relationship between androgen status and body composition is well established, the exact molecular basis by which androgens modulate skeletal muscle mass is not completely characterized but involves the repression of several atrogenes, induction of PI3K/AKT/mTOR signaling, and direct stimulation of muscle satellite cells (MUSCs) (Dubois *et al*, 2012). SARMs have clearly demonstrated the ability to attenuate orchiectomy- and glucocorticoid-mediated muscle loss in rodents (Jones *et al*, 2010), but seemingly at odds with their clinical development for cancer wasting, very few reports exist describing their efficacy in models of cancer-induced cachexia.

To date, several rodent models exhibiting cachexia in response to tumor burden have been described (Ballaro *et al*, 2016). These models vary in their cancer tissue of origin, kinetics of weight loss, and severity of cachectic decline (Chen *et al*, 2015; Toledo *et al*, 2016; Michaelis *et al*, 2017). One of the most widely used and best-characterized models involves the implantation of a carcinogen-induced colon cancer cell line (colon-26, C-26) originating from a female BALB/c mouse, which results in acute severe cachexia (Corbett *et al*, 1975; Talbert *et al*, 2014) We explored anabolic response to SARM therapy in the C-26 model and found SARM treatment had essentially no impact on muscle wasting associated with this common mouse model of cancer cachexia. In these mice, androgen-mediated skeletal muscle gene transcription was severely muted in the presence of a cachectic burden, and fully anabolic doses of SARM were unable to normalize tumor-mediated muscle E3-ligase expression (atrogin-1 and MuRF1) to effectively combat the catabolic decline driven by the C-26 tumor. Our objective was to better understand the failure of SARMs in the C-26 model in the hopes of gaining insight into the limitations of androgen therapy in cachectic cancer patients. To this end, we contrasted SARM treatment with an effective anti-cachectic histone deacetylase inhibitor (HDACi) regimen in the C-26 model.

We recently demonstrated the effectiveness of a novel class I/IIB HDAC inhibitor AR-42, currently under clinical evaluation in hematologic malignancies (Sborov *et al*, 2017) and solid tumors, as anti-cachexia therapy in the C-26 model (Tseng *et al*, 2015a). AR-42 administration in these mice spared body weight and was associated with improvements, but not complete rescue, of skeletal muscle mass relative to controls. Notably, AR-42 differed from other approved HDACis in its ability to fully suppress tumor-mediated atrogin-1 and MuRF1 induction and prolong survival in the C-26 model. Unlike androgens, AR-42 does not promote skeletal muscle hypertrophy in tumor-free animals, and AR-42's anti-cachectic efficacy was highly dependent on early initiation of treatment, suggesting AR-42's dramatic anti-cachectic efficacy in the C-26 model is primarily associated with its anti-catabolic effects (Tseng *et al*, 2015a). SARMs' established anabolic potential, but clear inability to attenuate tumor-driven wasting in the C-26 model, and AR-42's effects on tumor-mediated catabolic signaling, but apparent lack of anabolic effects on skeletal muscle, provided compelling rationale to explore co-administration of AR-42 with SARMs in well-established mouse models of cancer cachexia, the C-26, and Lewis lung carcinoma (LLC) models, as improved anti-cachectic therapy.

## Results

### AR-42 administration demonstrates limited anti-cachectic effects at a reduced 10 mg/kg dose level

We recently characterized the anti-cachectic effects of AR-42 (50 mg/kg via oral gavage every other day) in C-26 tumor-bearing mice (Tseng *et al*, 2015a). In tumor-free mice, this dose of AR-42 had no significant effects on lower limb skeletal muscle masses relative to controls but resulted in roughly a 50% reduction in epididymal adipose tissue masses. This dose also represented the maximally tolerated dose in mice, which was used to observe its anti-tumor effects in different xenograft tumor models. To better understand the disposition of AR-42 following oral administration in mice, we performed a limited pharmacokinetic study of single oral doses of 50, 20, and 10 mg/kg of AR-42 (Fig 1A). Plasma exposure following oral administration of 50 mg/kg was 74.3 μM*h (Appendix Fig S1A), which exceeded the well-tolerated plasma exposure in humans of 8.5 μM*h by 8.7-fold (Sborov *et al*, 2017). Consequently, we evaluated the anti-cachectic effects of lower doses of AR-42 (1–20 mg/kg) in a dose–response study in the C-26 model. Similar to six total 50 mg/kg doses (administered q2d), thirteen daily oral doses of 20 or 10 mg/kg AR-42 ameliorated C-26 tumor-mediated reductions in tumor-corrected body weight (Fig 1B) and gastrocnemius mass (Appendix Fig S1B), whereas lower doses were not effective. AR-42 readily distributed into gastrocnemius muscle tissue (Fig 1A), and at the minimally efficacious dose of 10 mg/kg, muscle concentrations remained above 700 nM for 4 h, consistent with the ability of 1 μM AR-42 to inhibit class I (72–95%) and IIb (86–100%) HDACs based on its *in vitro* HDAC inhibition profile (Appendix Fig S1C). The plasma exposure resulting from the 10 mg/kg dose (10.9 μM*h; Appendix Fig S1A) compares more favorably to well-tolerated exposures in patients while providing anti-cachectic efficacy and was therefore utilized in subsequent combination studies.

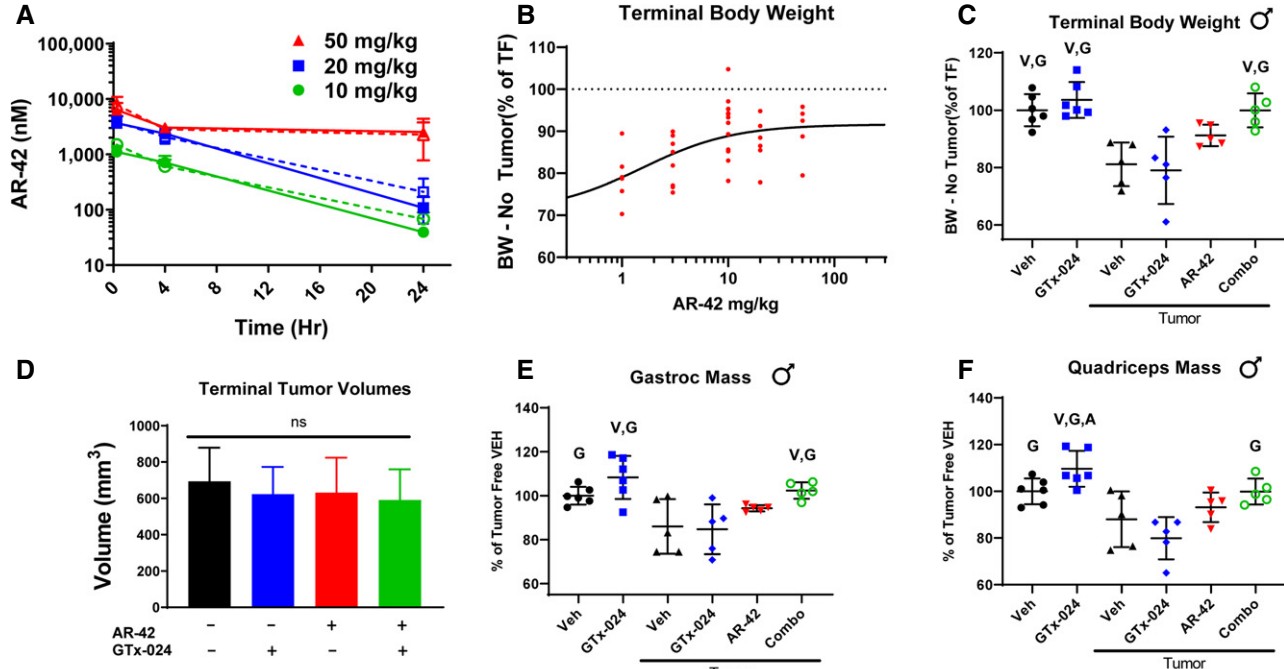

**Figure 1.** *In vivo* dose selection of AR-42 and evaluation of anti-cachectic effects of combination therapy with reduced dose AR-42 and SARM.

A  Single-dose AR-42 plasma and tissue pharmacokinetic study. Tumor-free CD2F1 mice were administered a single dose of 10, 20, or 50 mg/kg AR-42 (*n* = 3), and plasma (dashed) and gastrocnemius (solid) tissue were analyzed for AR-42 content at different times using LC-MS/MS analyses according to the Materials and Methods (mean ± SD).

B  AR-42 dose–response. Starting 6 days after C-26 cell injection, animals received vehicle or AR-42 orally at 1, 3, 10, or 20 mg/kg daily or 50 mg/kg every other day for 13 days. Individual animal terminal (day 18 post-injection) body weights corrected for tumor mass according to the Materials and Methods are compared to tumor-free controls. Groups: 1 mg/kg (*n* = 8), 3 mg/kg (*n* = 8), 10 mg/kg (*n* = 13), 20 mg/kg (*n* = 6), 50 mg/kg (*n* = 5). Dashed reference line (100%) represents the mean tumor-free control value, and solid line represents non-linear fit of dose–response data.

C–F  Study 1, tumor-bearing male mice receiving GTx-024 (15 mg/kg; *n* = 5), AR-42 (10 mg/kg; *n* = 5), combination (15 mg/kg GTx-024 and 10 mg/kg AR-42; *n* = 5), or vehicle (*n* = 5) and tumor-free mice receiving GTx-024 (15 mg/kg; *n* = 6) or vehicle (*n* = 6) were treated daily by oral gavage for 13 days starting 6 days post-injection of C-26 cells. (C): Terminal (day 18 post-injection) body weights corrected for tumor mass according to the Materials and Methods compared to tumor-free controls (mean ± SD). (D) Terminal tumor volumes (mean ± SD). (E) Terminal gastrocnemius (mean ± SD) and (F) quadriceps muscle mass (mean ± SD) compared to tumor-free controls.

Data information: V, G, A indicate significant differences versus tumor-bearing vehicle-treated, tumor-bearing GTx-024-treated, and tumor-bearing AR-42-treated groups, respectively. *P*-values provided in Appendix Table S1A–D, one-way ANOVA followed by Tukey's multiple comparison test. ns, not significant; BW, body weight; TF, tumor-free; Veh, vehicle.

**Combination GTx-024 and AR-42 administration results in improved anti-cachectic efficacy in the C-26 model**

To evaluate combined HDAC inhibition with SARM administration as improved anti-cachectic therapy, we designed a series of five studies combining AR-42 with androgen/SARM in mouse models of cancer wasting. In Study 1, similar to results from this and other laboratories (Bonetto *et al*, 2011; Tseng *et al*, 2015a), vehicle-treated tumor-bearing animals lost approximately 20% of their body weight prior to meeting euthanasia criteria (Fig 1C). Balanced tumor burdens among treatment groups (Fig 1D) drove severe tumor-induced weight loss in vehicle-treated controls and animals treated with 15 mg/kg GTx-024 (Fig 1C, 81.1 ± 7.6% and 79.0 ± 11.2% of tumor-free controls, respectively). Body weight loss was accompanied by parallel reductions in gastrocnemius and quadriceps masses in vehicle-treated controls (Fig 1E and F, 86.0 ± 12.4 and 88.0 ± 12.0%, relative to tumor-free controls, respectively) and GTx-024-treated mice (Fig 1E and F, 84 ± 11.3 and 80.0 ± 9.1%,

relative to tumor-free controls, respectively). SARM monotherapy had no apparent anti-cachectic efficacy in C-26 tumor-bearing mice. At this dose, GTx-024 was well tolerated in xenografted mice (Narayanan *et al*, 2014) and, in this study, did not cause body weight loss in tumor-free controls (Fig 1C). Furthermore, GTx-024 was reported to be fully anabolic at doses as low as 0.5 mg/kg/day in rodents (reported as S-22 in Kim *et al*) and compared favorably to the less potent structural analog S-23 (Jones *et al*, 2009), which prevented orchiectomy- and glucocorticoid-mediated wasting (Jones *et al*, 2010). A separate control study in tumor-free CD2F1 mice confirmed that, in our hands, 15 mg/kg GTx-024 was capable of increasing body weight, gastrocnemius and quadriceps mass, and grip strength in orchiectomized (ORX) mice relative to vehicle-treated ORX controls (Fig EV1). Importantly, GTx-024 suppressed serum-luteinizing hormone, a very well-characterized pharmacological effect of potent androgen administration (Nieschlag & Behre, 1998; Rommerts, 1998), demonstrating that GTx-024 administered to C-26 tumor-bearing mice in this study was active (Fig EV2A).

Similar to single-agent GTx-024 treatment, 10 mg/kg AR-42 failed to significantly improve body weight or hindlimb skeletal muscle masses over those of tumor-bearing vehicle-treated controls (Fig 1C, E and F, 91.2 ± 3.7%, 94.3 ± 1.4% and 93.2 ± 6.3% of tumor-free controls, for body, gastrocnemius, and quadriceps weights, respectively). In contrast to monotherapy, this combination exhibited a striking ability to consistently protect body weight (Fig 1C, 99.9 ± 5.9% of tumor-free controls, corrected for tumor weight) relative to either agent alone. Furthermore, the effects of combined therapy completely spared gastrocnemius (102.4 ± 3.8%) and quadriceps (99.9 ± 5.5%) mass relative to tumor-free controls (Fig 1E and F).

The effects of combined therapy on total body weight or amelioration of cachectic symptoms were not due to any overt impact on tumor burden as no significant differences in tumor volumes were apparent at the end of the study (Fig 1D). Food consumption was monitored to account for potential anti-anorexic effects of treatment on the cachectic sequela following C-26 cell inoculation. GTx-024-treated tumor-free control animals, as well as the combination-treated group, demonstrated small increases in per animal food consumption relative to other groups between days 14 and 16 (Fig EV2B), which are unlikely to account for differences in body weight apparent by study day 14 (treatment day 9), as well as end of study differences in skeletal muscle masses (Fig 1E and F).

These promising results prompted us to repeat the experiment with expanded animal numbers in male (Study 2) and female mice (Study 3). Our follow-up study in males resulted in significantly larger tumors and more aggressive wasting relative to Study 1, though no differences within treatment groups were apparent (Fig EV3A). As a result, the study was terminated a day early, after only 12 days of treatment. In accordance with this increased tumor burden, tumor-corrected body weights were more consistently reduced and to a larger degree in tumor-bearing controls (Fig 2A, 73.8 ± 4.2%, and Fig 1C, 81.1 ± 7.6% of tumor-free controls in Study 2 and 1, respectively), and larger losses in gastrocnemius (76.3 ± 8.1%) and quadriceps (69.1 ± 7.6%) mass relative to tumor-free controls were noted (Fig 2B and C). In the face of this more severe cachexia, combined AR-42 and GTx-024 administration significantly spared body weight (87.8 ± 5.9 of tumor-free controls), though not to the degree realized in Study 1 (Fig 1C vs. Fig 2A), while both AR-42 alone and the combination significantly improved gastrocnemius and quadriceps mass relative to GTx-024 monotherapy (Fig 2B and C). In Study 2, C-26 tumors were accompanied by reductions in forelimb grip strength (Fig 2D, 81.3 ± 18.9% of tumor-free controls), but, consistent with the improvements in hindlimb skeletal muscle mass, AR-42 alone and in combination with GTx-024 improved grip strength over vehicle-treated tumor-bearing controls. Unlike the adipose-sparing effect of the higher 50 mg/kg dose of AR-42 (Tseng et al, 2015a), the lower dose of 10 mg/kg had no impact on adipose tissue (Fig EV3B). As androgens are thought to actively prevent adipogenesis (Singh et al, 2003), SARM administration was not expected to protect against C-26 tumor-mediated fat losses. Indeed, no treatment-mediated effects on visceral abdominal adipose tissue were apparent (Fig EV3B). Critically, combination therapy-mediated improvements in cachectic sequelae were translated into improved survival (Fig 2E).

Male mice are generally considered more sensitive to C-26 tumor burden so the majority of reported C-26 studies utilize male animals

(Penna et al, 2016). Given that the C-26 cell line originated in a female mouse (Corbett et al, 1975) and the overlapping anabolic mechanisms of SARMs and male sex hormones (Jones et al, 2010), we sought to confirm the efficacy of combination therapy in C-26 tumor-bearing female mice (Study 3). Despite similar tumor burden to males (Appendix Fig S2), tumor-mediated losses in body weight, gastrocnemius, and quadriceps mass were reduced in females (Fig 3A–C, 88.9 ± 7.2%, 87.6 ± 8.1%, and 85.3 ± 6.7% of tumor-free controls, respectively). Similar to males, only combination therapy improved body weight (Fig 3A, 102.6 ± 5.5% of tumor-free controls). However, skeletal muscles in female mice were more responsive to combination therapy than those in males demonstrating improved gastrocnemius and quadriceps masses (Fig 3B and C, 104.6 ± 7.3% and 108.4 ± 5.8% of tumor-free controls) as compared to tumor-bearing mice treated with vehicle and either single-agent therapy. This increased skeletal muscle mass translated into improved grip strength following combination therapy (Fig 3D, 103.0 ± 13.2% of tumor-free controls).

## Anabolic resistance in the LLC model

The LLC model is another well-characterized rodent model of cancer wasting (Llovera et al, 1998; Chen et al, 2015) previously shown to be sensitive to the anti-cachectic effects of 50 mg/kg of AR-42 (Tseng et al, 2015a). To assess the broader applicability of our combination therapy and to determine its efficacy against cachexia caused by a tumor type evaluated clinically with SARM therapy (lung cancer), we treated male, LLC tumor-bearing mice (Study 5). This study revealed that 15 mg/kg GTx-024 stimulated LLC tumor growth (Fig EV4A), which confounded data interpretation requiring the identification of a reduced, fully anabolic dose of GTx-024 that did not stimulate tumor growth. Pharmacokinetic analyses of plasma GTx-024 levels following a single 15 mg/kg oral dose (Fig EV4B) suggested a 30-fold dose reduction (0.5 mg/kg) would result in an exposure of ~28.8 μg*h/ml. This exposure is more than sixfold higher than the exposure projected to result from a 0.5 mg/kg oral dose in male rats (Kim et al, 2013), which has been previously shown to be fully anabolic (Kim et al, 2005). Consequently, this reduced 0.5 mg/kg dose level of GTx-024 was used in a second LLC study in which it did not promote LLC tumor growth (Fig EV4C). Vehicle-treated, LLC tumor-bearing mice lost body weight (Fig EV4D, 89.2 ± 6.1% of tumor-free controls) and skeletal muscle mass (Fig EV4E, gastrocnemius, 79.7 ± 5.7% of tumor-free controls), but did not respond to 10 mg/kg AR-42, 0.5 mg/kg GTx-024, or combination therapy.

## Multiple androgens demonstrate improved anti-cachectic efficacy when combined with AR-42

To confirm that the improvement of GTx-024's anti-cachectic efficacy in the C-26 model by co-administration with AR-42 was not a drug-specific phenomenon, tumor-bearing animals were treated with the SARM TFM-4AS-1 (Schmidt et al, 2010) and the potent endogenous androgen dihydrotestosterone (DHT) alone and in combination with AR-42 (Study 4). Similar to the 15 mg/kg dose of GTx-024, TFM-4AS-1 was administered at a previously characterized fully anabolic dose (10 mg/kg), but, as a monotherapy, did not spare body weight (Fig 4A) or mass of gastrocnemius or quadriceps

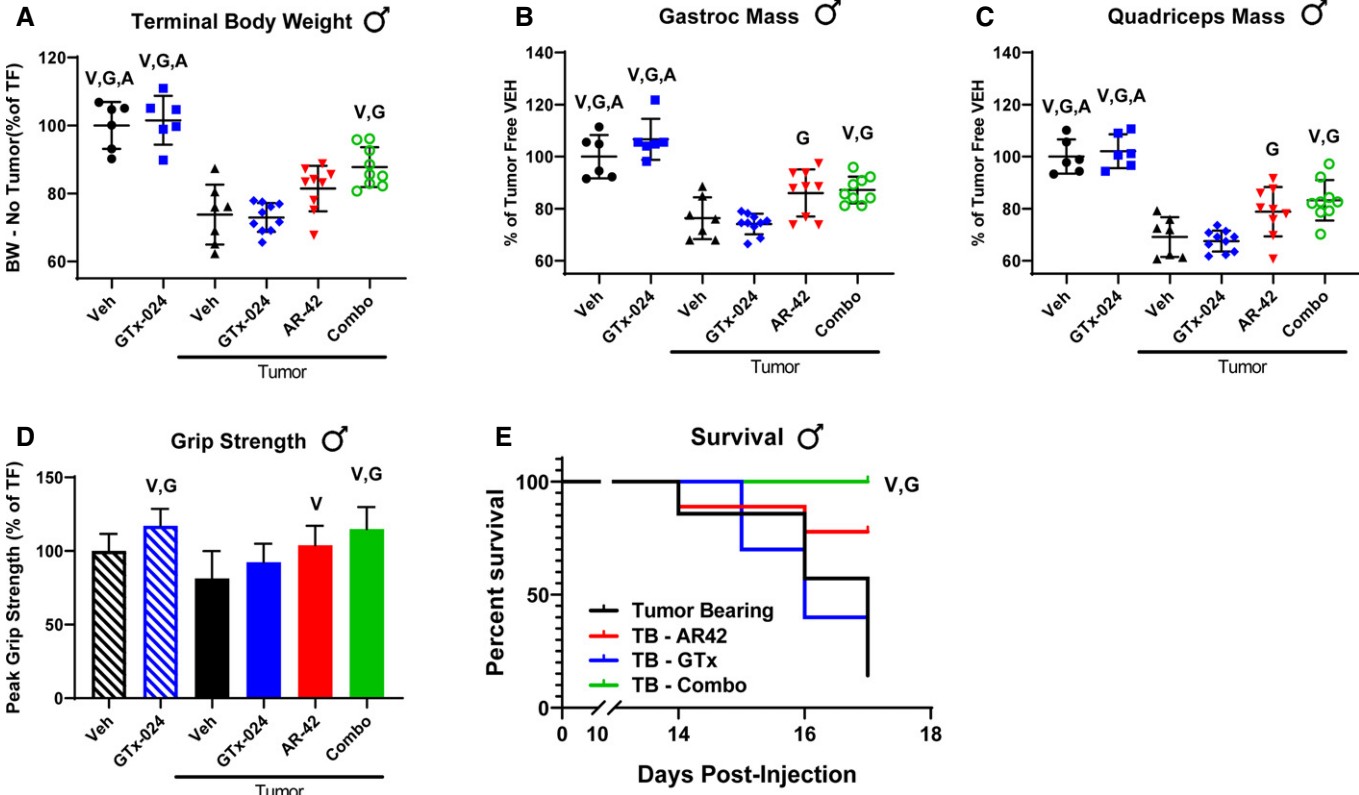

**Figure 2. Anti-cachectic effects of combination therapy in C-26 tumor-bearing mice.**

Study 2, tumor-bearing male mice receiving GTx-024 (15 mg/kg; *n* = 10), AR-42 (10 mg/kg; *n* = 9), combination (15 mg/kg GTx-024 and 10 mg/kg AR-42; *n* = 9), or vehicle (*n* = 7) and tumor-free male mice receiving GTx-024 (15 mg/kg; *n* = 6) or vehicle (*n* = 6) were treated daily by oral gavage for 12 days starting 6 days post-injection of C-26 cells.

A       Terminal body weights corrected for tumor mass according to the Materials and Methods compared to tumor-free controls (mean ± SD).
B–D     (B) Terminal gastrocnemius (mean ± SD) and (C) quadriceps muscle mass (mean ± SD) and (D) grip strength (mean ± SD) compared to tumor-free controls.
E       Survival analyses of Study 2 animals performed as outlined in the Materials and Methods.

Data information: (A–D) V, G, A indicate significant differences versus tumor-bearing vehicle-treated, tumor-bearing GTx-024-treated, and tumor-bearing AR-42-treated groups, respectively. *P*-values provided in Appendix Table S2A–D, one-way ANOVA followed by Tukey's multiple comparison test. (E) V, G indicate significant differences versus tumor-bearing vehicle-treated and tumor-bearing GTx-024-treated groups, respectively. *P* = 0.0005 for combo vs. V, *P* = 0.0064 for combo vs. G, log-rank (Mantel–Cox) test using Bonferroni-corrected threshold for multiple comparisons. BW, body weight; TF, tumor-free; TB, tumor-bearing; Veh, vehicle.

(Fig 4B and C). The DHT/AR-42 combination significantly improved body weights compared to vehicle-treated controls, as well as AR-42 and TFM-4AS treatment alone (Fig 4A, 100.6 ± 8.2% of tumor-free controls). Of note, tumor-bearing animals treated with DHT alone did not differ in initial tumor volumes (day 8), but after 8 days of DHT administration, tumor growth was significantly suppressed resulting in the exclusion of DHT-alone-treated animals from further analyses (Appendix Fig S3A). We determined that C-26 cells and tumor tissue express the androgen receptor (AR; Appendix Fig S3B), but did not detect direct effects of DHT treatment on C-26 cell viability capable of explaining reduced tumor volumes in the DHT monotherapy group (Appendix Fig S3C). Consistent with both Studies 1 and 2, improvements in body weight were not due to sparing adipose tissue as no treatment-mediated effects on adipose were apparent (Appendix Fig S4).

Similar to our other experiments, combination treatment-mediated improvements in body weight were translated to increased skeletal muscle masses as the DHT/AR-42 combination significantly

spared both gastrocnemius and quadriceps mass (93.7 ± 8.0 and 87.5 ± 6.1% of tumor-free controls, respectively). Tumor-mediated deficits in grip strength were not as apparent in this study, but the only treatment resulting in significantly improved grip strength was the combination of TFM-4AS-1 and AR-42, which increased muscle function over baseline (114.6 ± 16.6.1% of tumor-free controls) despite the presence of C-26 tumors.

## Effects of tumor burden and GTx-024/AR-42 treatment on the expression of AR and atrophy-related genes in skeletal muscle

Candidate gene expression analyses were performed on gastrocnemius tissue from Study 1 to characterize the effects of C-26 tumors and treatment with GTx-024, AR-42, or both agents on genes whose function has been previously associated with C-26 tumor-mediated wasting (Fig 5A). As expected for this model, the muscle-specific E3 ligases atrogin-1 (FBXO32) and MuRF-1(TRIM63) were induced in skeletal muscles of tumor-bearing animals (Bonetto *et al*, 2011;

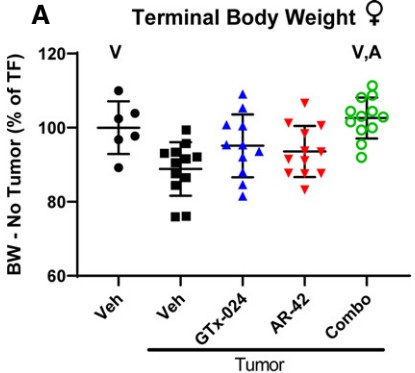

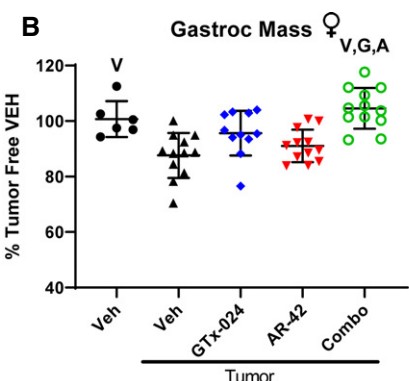

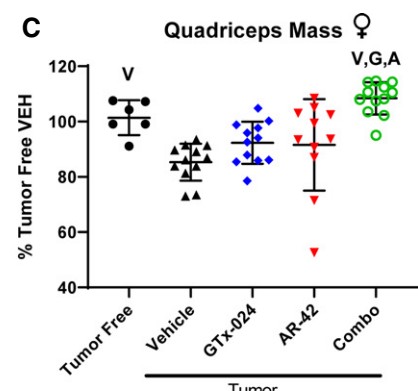

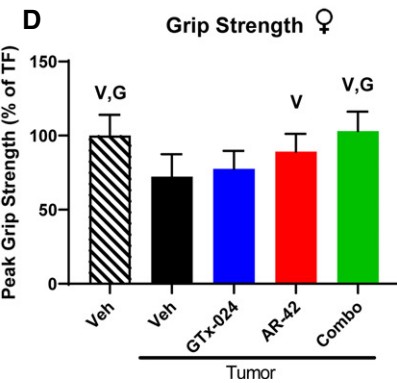

**Figure 3. Anti-cachectic effects of combination therapy in female C-26 tumor-bearing mice.**

Study 3, tumor-bearing female mice receiving GTx-024 (15 mg/kg; *n* = 12), AR-42 (10 mg/kg; *n* = 12), combination (15 mg/kg GTx-024 and 10 mg/kg AR-42; *n* = 12), or vehicle (*n* = 12) and tumor-free mice receiving vehicle (*n* = 6) were treated daily by oral gavage for 13 days starting 6 days post-injection of C-26 cells.

A Terminal body weights corrected for tumor mass according to the Materials and Methods compared to tumor-free controls (mean ± SD).

B–D (B) Terminal gastrocnemius (mean ± SD) and (C) quadriceps muscle mass (mean ± SD) and (D) grip strength (mean ± SD) compared to tumor-free controls.

Data information: V, G, A indicate significant differences versus tumor-bearing vehicle-treated, tumor-bearing GTx-024-treated, and tumor-bearing AR-42-treated groups, respectively. *P*-values provided in Appendix Table S3A–D, one-way ANOVA followed by Tukey's multiple comparison test. BW, body weight; TF, tumor-free; Veh, vehicle.

---

Tseng *et al*, 2015a) as was the STAT3 target gene and regulator of atrogin-1 and MuRF-1, CEBPδ (CEBPD; Silva *et al*, 2015). Consistent with the absence of any anti-cachectic effects of GTx-024 monotherapy, this treatment had no significant impact on atrogin-1, MuRF-1, or CEBPδ expression. Ten milligram/kilogram AR-42 alone and in combination with GTx-024 significantly reduced the expression of each atrogene relative to tumor-bearing controls, returning them to near baseline levels. AR-42's effects on E3 ligase expression were consistent with results from animals receiving the higher dose of 50 mg/kg (Tseng *et al*, 2015a), further supporting the importance of AR-42's ability to reverse induction of these key enzymes to its overall anti-cachectic efficacy.

To determine the effect of tumor burden and treatment on AR levels in skeletal muscle, which could influence response to androgen therapy, gastrocnemius AR levels were characterized. Neither tumor nor treatment had a significant impact on AR mRNA (Fig 5B). AR protein expression in gastrocnemius was low in tumor-free controls and increased in response to GTx-024 administration irrespective of tumor burden (Appendix Fig S5), consistent with androgen agonist binding and stabilization of the AR (Kemppainen *et al*,

1992). In contrast, AR-42 treatment did not have a marked impact on AR expression.

### Anti-cachectic efficacy of AR-42 is associated with STAT3 inhibition but not general immune suppression

Previously reported ingenuity pathway analyses of AR-42-regulated genes in gastrocnemius muscle revealed that 66 genes associated with muscle disease or function were significantly regulated by AR-42 relative to C-26 tumor-bearing vehicle-treated controls (Tseng *et al*, 2015a). In an effort to enrich previously reported differentially regulated genes (*n* = 548) for transcripts critical to the anti-cachectic efficacy of AR-42, these data were intersected with previously published differentially regulated genes from the quadriceps of moderate and severely wasted C-26 tumor-bearing mice (Bonetto *et al*, 2011; Data ref: Zimmers *et al*, 2011) (*n* = 700, Appendix Fig S6A). Using this approach, the likely biological relevance of the 147 overlapping genes is increased when it is considered that these transcripts represent genes regulated by AR-42 that are associated with C-26-induced wasting from two different muscles (gastrocnemius

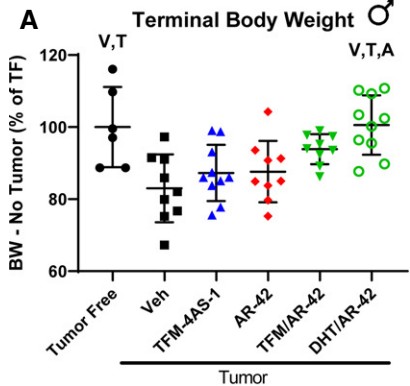

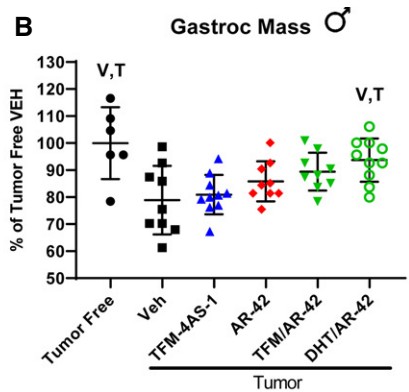

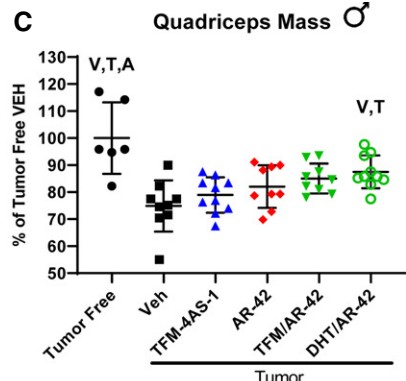

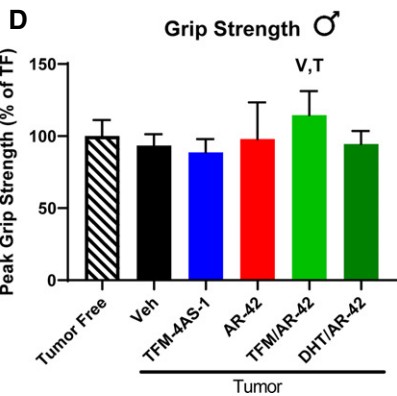

**Figure 4. Multiple androgens improved anti-cachectic efficacy when combined with AR-42.**

Study 4, tumor-bearing male mice receiving AR-42 (10 mg/kg, oral gavage; *n* = 9), TFM-4AS-1 (10 mg/kg, subcutaneous; *n* = 10), combination of AR-42 and DHT (10 mg/kg oral gavage and 3 mg/kg subcutaneous, respectively; *n* = 10), combination of AR-42 and TFM-4AS-1 (10 mg/kg, both, *n* = 9), or vehicle (*n* = 6) and tumor-free mice receiving vehicle (*n* = 6) were treated daily for 12 days starting 6 days after cell injection.

A    Terminal body weights corrected for tumor mass according to the Materials and Methods compared to tumor-free controls (mean ± SD).

B–D   (B) Terminal gastrocnemius (mean ± SD) and (C) quadriceps muscle mass (mean ± SD) and (D) grip strength (mean ± SD) compared to tumor-free controls.

Data information: V, T, A indicate significant differences versus tumor-bearing vehicle-treated, tumor-bearing TFM-4AS-1-treated, and tumor-bearing AR-42-treated groups, respectively. *P*-values provided in Appendix Table S4A–D, Tukey's multiple comparison test. BW, body weight; TF, tumor-free; Veh, vehicle.

and quadriceps), detected by two different technologies (RNA-seq and microarray), and reported by two different research laboratories. Pathway analyses performed on this pool of 147 genes revealed IL-6 signaling and immune system pathways, along with other gene sets regulated subsequent to cytokine stimulation, implicating AR-42's effects on cytokine and immune signaling in its anti-cachectic efficacy (Appendix Fig S6B).

In agreement with the present pathway analyses, we previously reported that the higher 50 mg/kg dose of AR-42 reduced serum IL-6 levels, as well as gastrocnemius IL-6 receptor mRNA abundance in tumor-bearing mice, suggesting AR-42's efficacy may be related to its suppression of systemic IL-6 activation, which is thought to drive muscle wasting in the C-26 model (Tseng *et al*, 2015a). In this study, the impact of C-26 tumor burden and treatment with AR-42, GTx-024, or combination therapy on a panel of circulating cytokines, including IL-6, was assessed (Table 1, Appendix Table S13). Although inter-animal variability among these cytokines was large potentially limiting meaningful interpretation of the data, statistically significant increases in multiple pro-cachectic factors,

including G-CSF, IL-6, and LIF, were detected in the presence of C-26 tumors, consistent with our previous report (Tseng *et al*, 2015a). Given this variability, smaller, treatment-mediated changes in circulating cytokines were not readily determined without a large expansion of animal numbers. Taking this limitation into account, the 10 mg/kg AR-42 dose did not have a large impact on IL-6 family cytokine levels (i.e., IL-6 or LIF) alone or in combination with GTx-024. Furthermore, 10 mg/kg AR-42 monotherapy did not significantly reduce circulating levels of any evaluated cytokine. An ELISA analysis confirmed our findings that 10 mg/kg AR-42 treatment did not have a strong impact on the variable circulating IL-6 levels associated with C-26 tumor burden (Fig 6A), and further demonstrated serum IL-6 levels were not associated with effects on body weight in treated, C-26 tumor-bearing mice at sacrifice ($r^2$ = 0.018, *P* = 0.48).

When clear effects on circulating cytokines were not apparent, we hypothesized AR-42 might be acting downstream of the IL-6 receptor on critical mediators of cytokine signaling. One well-characterized effector of cytokine-induced signaling shown to be central to tumor-induced wasting in a number of models is signal

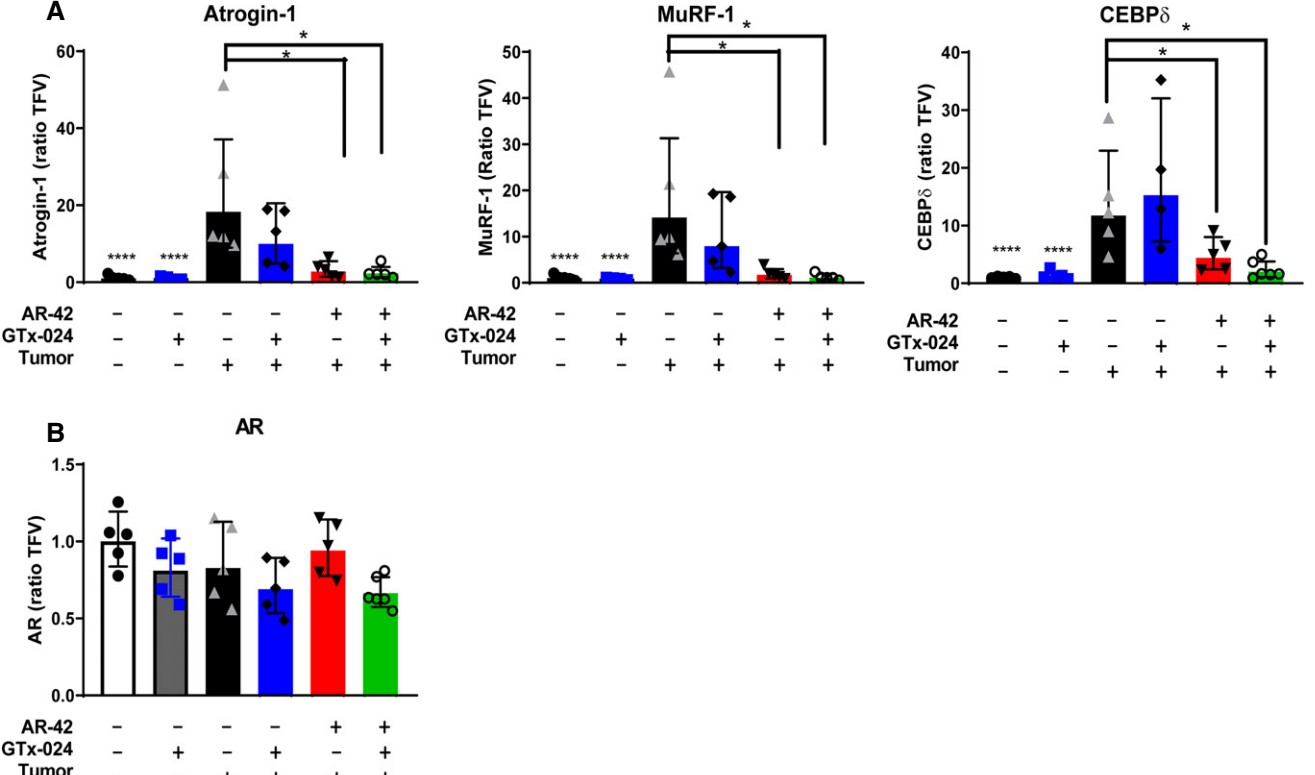

**Figure 5. Effects of tumor burden and GTx-024/AR-42 treatment on the expression of AR and atrophy-related genes in skeletal muscle.**

A   Gene expression of multiple cachexia-associated markers in gastrocnemius muscles of individual animals from Study 1 ($n = 5$ for tumor-bearing groups; $n = 6$ for tumor-free groups). Expression was determined by qRT–PCR and presented as described in the Materials and Methods (geometric mean ± geometric STD).

B   Androgen receptor (AR) mRNA expression in gastrocnemius muscles from Study 1.

Data information: P-values provided in Appendix Table S5A and B. Dunnett's multiple comparison test. CEBPδ, $n = 4$, insufficient sample to analyze all tumor-bearing GTx-024-treated animals.

**Table 1. Serum cytokine panel. Multiplex analysis of diverse serum cytokines[a] in terminal samples from Study 2[b].**

|  | Tumor-free | | C-26 Tumor-bearing | | | |
|---|---|---|---|---|---|---|
|  | Vehicle | GTx-024 | Vehicle | GTx-024 | AR-42 | Combo |
| G-CSF | 248.66 ± 64.60* | 338.39 ± 71.70* | 12164.11 ± 18944.48 | 2446.63 ± 1625.70* | 2782.18 ± 2191.30 | 1674.20 ± 1160.74* |
| GM-CSF | 18.71 ± 5.56 | 13.27 ± 4.62* | 21.92 ± 5.36 | 17.35 ± 4.33 | 18.70 ± 3.77 | 20.58 ± 5.40 |
| IL-6 | 3.35 ± 1.51* | 2.45 ± 1.31* | 537.66 ± 417.18 | 397.54 ± 341.43 | 256.59 ± 183.1 | 448.16 ± 294.52 |
| IL-17 | 3.04 ± 2.26 | 5.01 ± 1.18* | 1.30 ± 0.57 | 1.75 ± 1.28 | 1.82 ± 0.85 | 2.11 ± 1.21 |
| IP-10 | 162.64 ± 43.04 | 145.68 ± 48.83* | 238.29 ± 124.78 | 154.76 ± 17.98* | 215.35 ± 52.46 | 227.77 ± 45.23 |
| KC | 65.92 ± 26.47 | 90.02 ± 17.69 | 326.10 ± 215.79 | 288.89 ± 154.46 | 363.38 ± 200.65 | 1094.01 ± 528.53* |
| LIF | 2.03 ± 2.17* | 2.50 ± 2.34 | 24.51 ± 11.26 | 45.26 ± 21.57* | 15.79 ± 5.15 | 28.08 ± 21.16 |
| LIX | 3254.87 ± 3474.12 | 1316.67 ± 1662.66 | 4211.17 ± 3120.65 | 5234.39 ± 4771.34 | 2515.38 ± 3119.67 | 1663.11 ± 1732.77 |
| M-CSF | 47.72 ± 27.44* | 27.23 ± 10.09 | 23.63 ± 8.29 | 22.23 ± 9.45 | 20.21 ± 4.63 | 21.00 ± 4.19 |

G-CSF, granulocyte colony-stimulating factor; GM-CSF, granulocyte macrophage colony-stimulating factor; IL-17, interleukin-17; IL-6, interleukin-6; IP-10, interferon gamma-induced protein 10; KC, chemokine (C-X-C motif) ligand 1; LIF, leukemia inhibitory factor; M-CSF, macrophage colony-stimulating factor.
[a]Presented cytokines (pg/ml; mean ± SD) are limited to those showing significant differences from tumor-bearing vehicle-treated controls (*$P < 0.05$, one-way ANOVA followed by Dunnett's multiple comparison test). Complete cytokine data are presented in Appendix Table S5.
[b]Tumor-free groups, $n = 6$; tumor-bearing groups: vehicle-treated ($n = 7$), GTx-024-treated ($n = 10$), AR-42-treated ($n = 9$), and combination-treated ($n = 9$).

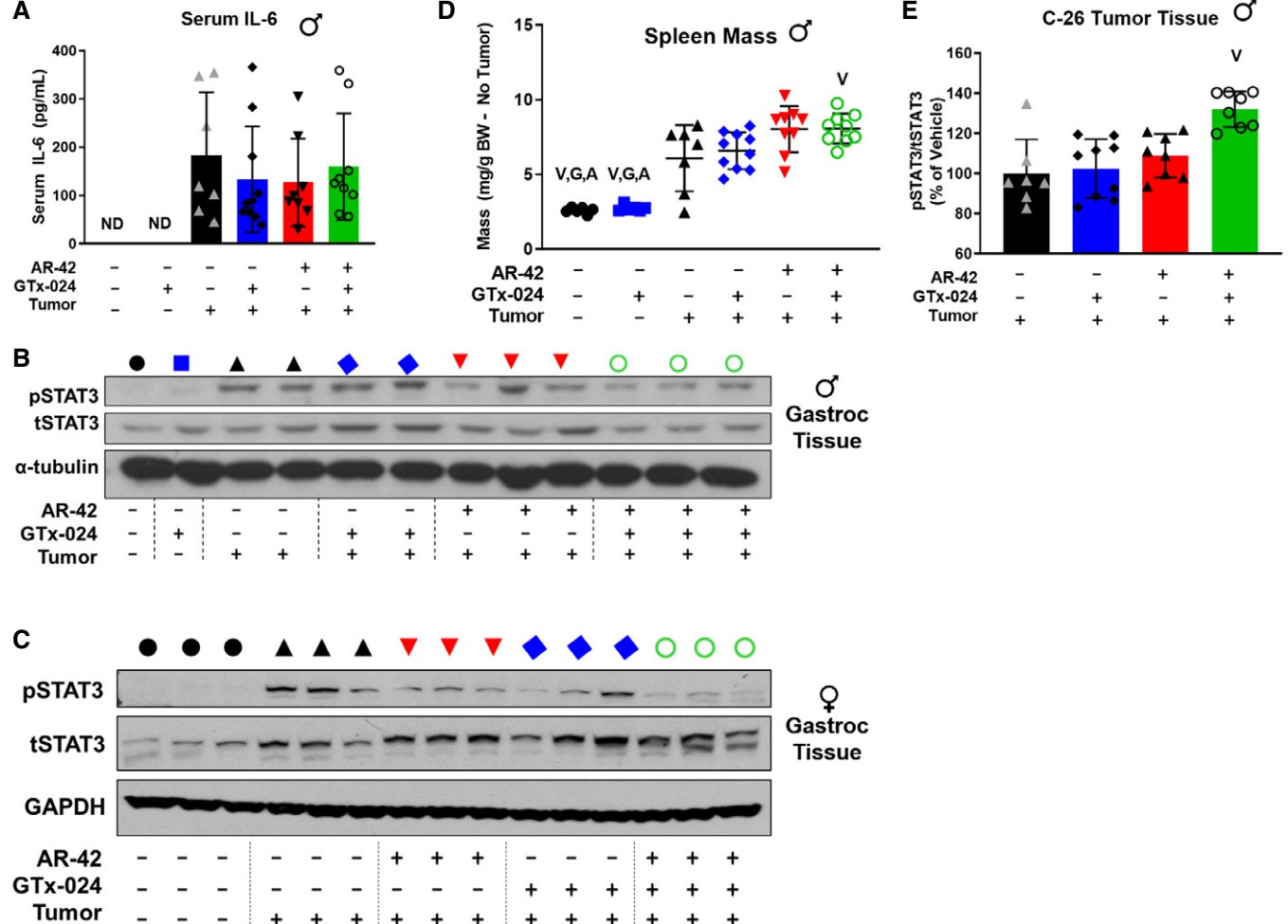

**Figure 6.  Anti-cachectic efficacy of AR-42 is associated with STAT3 inhibition but not general immune suppression.**

A    ELISA analysis of serum IL-6 levels in terminal samples from Study 2. Tumor-free groups ($n = 6$) and tumor-bearing groups receiving vehicle ($n = 7$), GTx-024 ($n = 10$), AR-42 ($n = 8$), or combination ($n = 9$). ND, not detected.

B, C  Phospho(p)STAT3 (Y705)/total (t)STAT3 Western blot analysis of gastrocnemius tissues from individual animals treated in Study 1 (B) and Study 3 (C). Black circle—tumor-free, blue square—tumor-free/GTx-024, black triangle—tumor-bearing, blue diamond—tumor/GTx-024, red triangle—tumor/AR-42, green circle—tumor/combo.

D    Spleen weights normalized to tumor-corrected terminal body weights of mice from Study 2. Tumor-free groups ($n = 6$), tumor-bearing vehicle ($n = 7$), and tumor-bearing groups receiving GTx-024 ($n = 10$), AR-42 ($n = 9$), or combination ($n = 9$).

E    ELISA analysis of pSTAT3(Y705)/tSTAT3 within C-26 tumors from Study 2. Tumor-bearing vehicle-treated ($n = 7$) and tumor-bearing groups receiving GTx-024, AR-42, or combination ($n = 8$).

Data information: V, G, A indicate significant differences versus tumor-bearing vehicle-treated, tumor-bearing GTx-024-treated, and tumor-bearing AR-42-treated groups, respectively. *P*-values provided in Appendix Table S6, one-way ANOVA followed by Tukey's multiple comparison test (mean ± SD). Panel (E), one sample from each of the treatment groups was not available for analyses. BW, body weight.

Source data are available online for this figure.

transducer and activator of transcription (STAT)3 (Bonetto *et al*, 2011; White *et al*, 2013b). Notably, STAT3 activation is associated with the severity of wasting in both the C-26 and $Apc^{min/+}$ models of cancer cachexia, and AR-42 was previously shown to suppress the IL-6/GP130/STAT3 signaling axis in multiple myeloma cells (Zhang *et al*, 2011). Thus, we evaluated AR-42's effects on phospho-STAT3 (pSTAT3) in gastrocnemius muscle from C-26 tumor-bearing animals (Fig 6B and C, Appendix Fig S7). In males, as expected, the presence of the C-26 tumor resulted in increased

pSTAT3 abundance. GTx-024 treatment had no apparent effect on pSTAT3, consistent with its inability to spare body weight or hindlimb skeletal muscle mass as a monotherapy (Fig 6B). AR-42 monotherapy reduced pSTAT3 but not equally in all animals, whereas the combination treatment exhibited the most consistent suppression, concordant with its marked anti-cachectic efficacy. Furthermore, treatment-mediated effects on the well-characterized STAT3 target gene CEBPδ (Silva *et al*, 2015) closely paralleled those on STAT3 activation (Fig 5A). Female mice demonstrated a similar

induction in gastrocnemius pSTAT3 levels in response to C-26 tumor burden (Fig 6C). However, both GTx-024 and AR-42 monotherapy resulted in reduced levels of pSTAT3 activation. Combination-treated female mice had the lowest levels of pSTAT3 activation among tumor-bearing mice, similar to effects seen in males.

In addition to skeletal muscle STAT3 activation, C-26 tumor-bearing mice exhibit splenomegaly as a result of increased systemic inflammation (Aulino et al, 2010). C-26 tumor-bearing animals in both Studies 1 and 2 demonstrated large increases in spleen mass across all treatment groups relative to tumor-free controls (Figs 6D and EV2C, respectively). Similar to findings with 50 mg/kg AR-42 (Tseng et al, 2015a), spleen mass was either unchanged or slightly increased by AR-42 alone or in combination with GTx-024. As a gross measure of the systemic effects of treatment on immune function, these spleen mass results suggest AR-42 is not generally immunosuppressive and its activity is distinct from inhibitors of the JAK/STAT pathway in this context (Mesa et al, 2012). Unlike in gastrocnemius tissue, AR-42 treatment did not significantly suppress pSTAT3 signaling within the C-26 tumors themselves (Fig 6E). Taken together, these multiple lines of evidence suggest that the anti-cachectic efficacy of AR-42 involves the inhibition of the IL-6/GP130/STAT3 axis in skeletal muscle tissue, but not systemic suppression of IL-6 or general immune signaling.

**Transcriptomic analyses of AR-42's anti-cachectic effects in skeletal muscle**

To further characterize AR-42's anti-cachectic effects at the reduced dose of 10 mg/kg, RNA-seq analyses were performed on all gastrocnemius tissues from Study 1 (Fig 1E). This resulted in 31 evaluable samples across treatment groups (Appendix Fig S8) after removal of two samples due to insufficient sequencing yield/quality. We detected 4,579 differentially expressed genes (DEGs; FDR < 0.1) in cachectic versus control muscle, whereas treatment of cachectic mice with GTx-024 or AR-42 alone resulted in 5,561 and 723 DEGs, respectively (Fig 7A, Appendix Fig S9A and B). Given the ability of HDAC inhibitors and androgens to modulate transcription, initial functional analyses were focused on curated *Mus musculus* transcription factor (TF) target gene sets, and revealed multiple over-represented TF targets in cachectic versus control muscle (Fig 7B). STAT3 and activation of transcription-1 (ATF1) gene sets were each represented twice in the top ten pathways following gene set enrichment analysis (GSEA) supporting their potential relevance in cachectic signaling. The two STAT3 target gene sets were combined, and GSEA was repeated with the combined set for all treatment groups. In contrast to pSTAT3 activation (Fig 6B and C), this analysis demonstrated the inability of any treatment in tumor-bearing mice to significantly limit the importance of STAT3 target gene regulation relative to cachectic controls (Fig 7C and Appendix Fig S10). However, when analysis is focused on individual genes within the combined set that are differentially expressed in at least one comparison, clear cachexia-dependent regulation is apparent that responds only to AR-42 treatment (Fig 7D). A similar analysis with combined ATF-1 data sets revealed the ability of AR-42, but not GTx-024 treatment, to significantly impact ATF-1 target gene regulation in tumor-bearing mice implicating AR-42's ability to modulate ATF-1 activation in its anti-cachectic efficacy (Fig 7E and

Appendix Fig S11). Of note, STAT3 and CEPBδ are among the differentially expressed ATF-1 target genes induced by cachexia that respond only to AR-42 treatment (Fig 7F).

We further evaluated the expression of genes within the IL-6 pathway as IL-6-mediated STAT3 target gene regulation is well characterized in the C-26 model (Bonetto et al, 2011), and IL-6-mediated increases in skeletal muscle cyclic AMP (cAMP), a primary driver of ATF-1 activation (Rehfuss et al, 1991), have also been reported (Kelly et al, 2009). Unlike circulating IL-6 cytokine, IL-6 mRNA in gastrocnemius muscle was not induced by cachexia, nor was it modulated by any treatment (Fig 7G). However, expression of both IL-6 receptor (IL-6RA) and the key effector GP130 was elevated in cachectic mice and required AR-42 (IL-6RA) or combination treatment (GP130) to restore expression to non-cachectic control levels.

Considerable overlap exists between the transcriptomes of cachectic gastrocnemius muscles from mice treated with 10 or 50 mg/kg AR-42 such that high fold-change DEGs identified by Tseng et al and in the current study are all regulated in the same direction (n = 209; Appendix Fig S12A and B). Similar to previous analyses (Appendix Fig S6), functional interrogation of the genes within this overlap further supports the importance of AR-42's ability to modulate immune and extracellular matrix signaling in eliciting its anti-cachectic effects (Appendix Fig S12C). Taken together, these findings support the ability of the reduced 10 mg/kg dose of AR-42 to generate anti-cachectic effects by reducing pro-cachectic IL-6RA/GP130/STAT3 signaling in skeletal muscle.

**Transcriptomic analyses of GTx-024's anabolic effects in skeletal muscle**

To better understand GTx-024's contribution to the efficacy apparent in combination-treated mice, the transcriptome of combination-treated gastrocnemius muscle was compared to cachectic controls revealing 2,026 DEGs (or 50.6% of all DEGs) unique to combination-treated muscle and not solely attributable to AR-42 treatment (Fig 8A). We hypothesized that GTx-024-mediated anabolic signaling detectable in GTx-024-treated tumor-free controls would be diminished in cachectic, tumor-bearing GTx-024-treated animals in the absence of AR-42. Though very few DEGs were apparent in GTx-024-treated tumor-free controls (n = 27; Appendix Fig S13), GSEA focused on TF pathways revealed abundant coordinated signaling with regulation of β-catenin (CTNNB1) target genes providing the most significant overlap (FDR < 1e-5; Fig 8B). Coordinate regulation of β-catenin target genes was not apparent in cachectic controls or following GTx-024 or AR-42 monotherapy, but was again among the most prominent pathways detected by GSEA in combination-treated mice (FDR < 1e-5; Fig 8C). GSEA plots demonstrate a robust pattern of GTx-024-mediated activation of β-catenin target genes requiring AR-42 co-administration in cachectic mice (Fig 8D, *leftmost panel compared to rightmost panel*). Analysis of overlap of the leading edge genes revealed a large number of CTNNB1 target genes regulated by both GTx-024 and cachexia versus tumor-free controls but in different directions (n = 49 *middle*, 17 *bottom left*; Appendix Fig S14A). Many fewer leading edge genes were regulated by AR-42 monotherapy but also in an opposite direction to GTx-024 (n = 2 *middle*, 23 *top middle*; Appendix Fig S14B). However, combined therapy results in a larger leading edge gene set overlap that is regulated in a similar direction

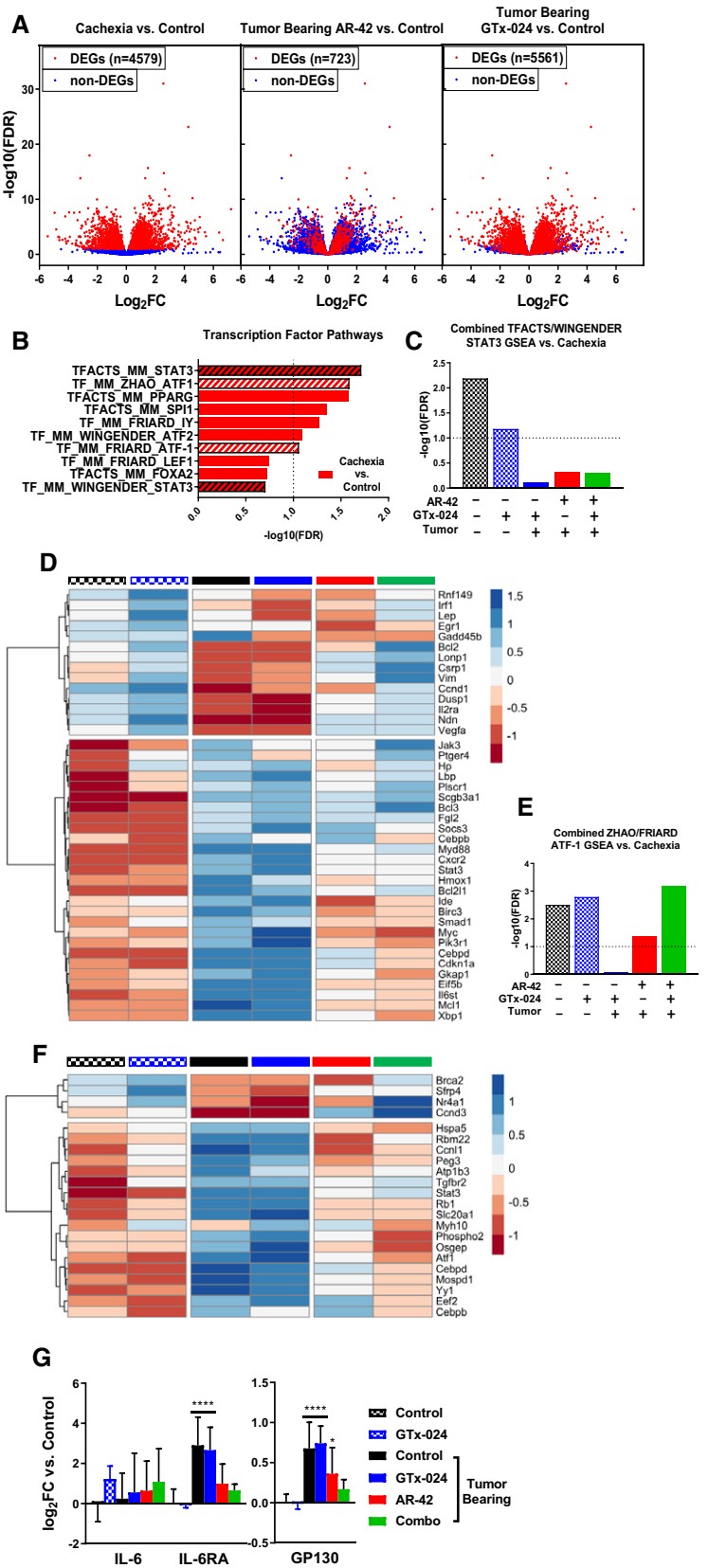

**Figure 7.**

◀

**Figure 7.  Transcriptomic analyses of AR-42's anti-cachectic effects in skeletal muscle.**

A   Effect of GTx-024 and AR-42 monotherapies on cachexia-related differentially regulated genes (DEGs) from RNA-seq analyses of Study 1 gastrocnemius muscles. All three panels consist of individual genes plotted with respect to their log2 fold-change and $-\log 10$ Benjamini–Hochberg-adjusted $P$-values from the comparison of cachexia vs tumor-free controls. Colors of the points reflect the DEG status of each gene for the given comparison.

B   Results from transcription factor pathway-focused GSEA of tumor-bearing (cachexia) versus tumor-free control transcriptomes. STAT3 and ATF-1 gene sets used for subsequent combined analyses are hatched.

C   Significance values from GSEA using combined STAT3 gene sets identified in B. Each treatment group is compared to tumor-bearing control (cachexia) transcriptomes.

D   Heatmap of DEGs within the combined STAT3 gene sets representing mean z-scores calculated from normalized RNA-seq count data. Tumor-free control (black checkered), GTx-024-treated tumor-free (blue checkered), tumor-bearing control (black), GTx-024-treated tumor-bearing (blue), AR-42-treated tumor-bearing (red), and combination-treated tumor-bearing (green) groups.

E   Results of GSEA using combined ATF-1 gene sets identified in (B) across treatment groups versus tumor-bearing control (cachexia) transcriptomes.

F   Heatmap of DEGs within the combined ATF-1 gene sets (mean z-score). Treatment groups are as in (D).

G   mRNA expression of mediators of IL-6 signaling upstream of STAT3. Data presented as mean ± SD of per animal log-transformed fold-change ($\log_2$FC) values versus tumor-free controls. Groups: Tumor-free control ($n = 6$), tumor-free GTx-024-treated ($n = 5$), and tumor-bearing groups receiving vehicle ($n = 4$), GTx-024 ($n = 5$), AR-42 ($n = 5$), or combination ($n = 6$). *$P < 0.1$, ****$P < 0.001$ based on Benjamini–Hochberg-adjusted $P$-values from DESeq2. Exact $P$-values are provided in Appendix Table S7.

to GTx-024 monotherapy ($n = 29$ *middle*, 37 of 47 *top middle*; Appendix Fig S14C). This pattern of β-catenin target gene regulation is also apparent when DEGs within the TFACTS_CTNNB1 gene set are visualized across treatment groups (Appendix Fig S15).

Expression of the canonical skeletal muscle WNT agonist Wnt5a (WNT5A), canonical WNT receptor Fzd1 (FZD1), and β-catenin itself (CTNNB1) was all reduced with C-26 tumor burden, whereas the negative regulator of β-catenin, GSK3B, was up-regulated (Fig 8E). In each case, GTx-024 monotherapy in tumor-bearing mice failed to restore expression to tumor-free control levels. However, with the exception of β-catenin, AR-42 treatment effectively prevented tumor-induced regulation. Furthermore, combination treatment alone restored β-catenin and the well-characterized β-catenin target gene cyclin D1 (CCND1) (Shtutman *et al*, 1999) expression to tumor-free control levels. Taken together, these data provide strong support for: (i) the dependence of GTx-024's anabolic effects in skeletal muscle on functional WNT/β-catenin signaling; (ii) C-26 tumor burden's ability to disrupt WNT/β-catenin signaling in skeletal muscle; and (iii) AR-42's ability to restore WNT/β-catenin responsiveness to treatment with GTx-024.

## Discussion

Our initial objective was to determine cachectic signaling occurring in common mouse models of cancer wasting that limited the effectiveness of anabolic SARM therapy. Upon determining that catabolic E3-ligase expression in skeletal muscle was sensitive to androgen administration in hypogonadism- and dexamethasone-induced wasting (Jones *et al*, 2010; White *et al*, 2013a), but not in wasting driven by C-26 tumor burden, we evaluated potential drivers of cachectic signaling in skeletal muscle and their sensitivity to androgens. Our familiarity with the anti-cachectic effects of AR-42 administration (Tseng *et al*, 2015a) led us to more closely consider the mechanisms of anti-cachectic HDACi administration and whether a primarily anti-catabolic agent could be combined with a well-characterized anabolic agent as an improved anti-cachexia therapy. We showed that common mouse models of cancer cachexia are refractory to anabolic androgen administration, but, in some cases, anti-cachectic efficacy can be markedly improved by combined treatment with the HDACi AR-42 (Figs 1–4). The successes of these combination treatments provided a unique

opportunity to better understand cachectic drivers in skeletal muscle sensitive to AR-42 treatment that may be limiting anabolic androgen signaling in cachectic skeletal muscle. To this end, one of our key findings was increased insight into the anti-cachectic mechanism of AR-42.

### Anti-cachectic mechanism of AR-42

AR-42 is currently under clinical evaluation as a direct anti-tumor agent (NCT02282917, NCT02795819, NCT02569320). Driven by tolerability concerns from recent clinical experience, we evaluated a fivefold AR-42 dose reduction and found that elements of anti-cachectic efficacy were retained across multiple studies. This allowed us to focus on key aspects of AR-42's pharmacologic activity that contribute to its anti-cachectic effects. We report that STAT3 activation was sensitive to AR-42 treatment in cachectic skeletal muscle (Fig 6B and C), which is consistent with the demonstration by Seto *et al* (2015) that IL-6 family cytokine signaling through STAT3 is a critical mediator of C-26-induced wasting. These findings are in agreement with our transcriptomic analyses, which substantiated both STAT3 and ATF-1 transcriptional programs as cachectic drivers (Fig 7). AR-42 treatment reduced IL-6RA and GP130 mRNA (Fig 7G) similar to reports of AR-42 activity in multiple myeloma cells (Zhang *et al*, 2011) and the activity of pan-HDACis in naïve CD4[+] T cells (Glauben *et al*, 2014). Tissue-specific HDACi-mediated muting of IL-6R and/or GP130 induction following cachectic challenge provides a plausible mechanism for the reversal of IL-6 family cytokine-driven ATF-1/STAT3 transcription we detected in the absence of broader systemic immune effects. Determining the precise mechanism by which treatment with AR-42, but not other HDACis (Tseng *et al*, 2015a), mediates anti-cachectic efficacy will require further study, but our data warrant continued evaluation of AR-42 as an anti-cachectic agent.

### Impact of cachectic tumor burden on androgen signaling

Another key finding of this report is the extent of resistance to anabolic androgen administration in the C-26 and LLC models of cancer cachexia. In the C-26 model, we utilized fully anabolic doses of two SARMS (GTx-024, TFM-4AS-1) and a potent steroidal androgen (DHT), which resulted in no detectable efficacy. This occurred

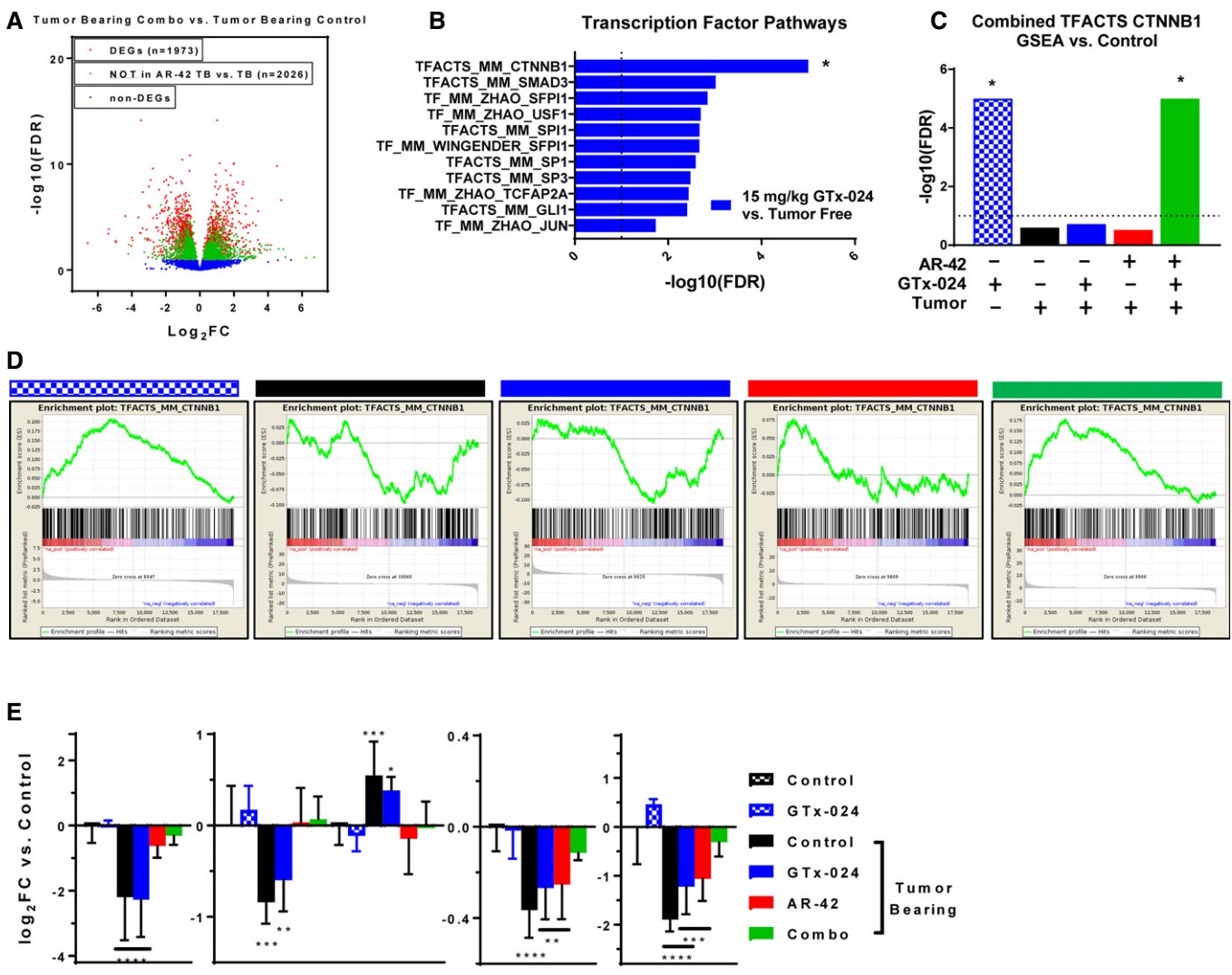

**Figure 8. Transcriptomic analyses of GTx-024's anabolic effects in skeletal muscle.**

A   Volcano plot from RNA-seq analyses of Study 1 gastrocnemius muscles for tumor-bearing combination-treated mice versus tumor-bearing controls. Genes not differentially expressed in this comparison are indicated in blue. The remaining genes (red and green) are DEGs in the combination-treated versus tumor-bearing control comparison. The green coloring indicates the subset of these DEGs that are not also differentially expressed in the comparison of AR-42-treated tumor-bearing mice versus tumor-bearing controls, suggesting these genes are responsive to only the combination therapy. Log2-transformed fold change (FC) in expression is plotted on the x-axis, and -log10-transformed Benjamini–Hochberg-adjusted P-values are plotted on the y-axis.

B   Significance values from transcription factor pathway-focused GSEA of GTx-024-treated tumor-free versus tumor-free control transcriptomes. *FDR < 1e-5 was determined for the CTNNB1 gene set and was set to 1e-5 for the plot.

C   Significance values from GSEA using combined CTNNB1 gene sets. Each treatment group was compared to tumor-free control transcriptomes. *FDR < 1e-5 determined, set to 1e-5 for plot.

D   Enrichment plots from GSEA of the CTNNB1 gene set for each treatment group versus tumor-free control comparisons. GTx-024-treated tumor-free (blue checkered), tumor-bearing control (black), GTx-024-treated tumor-bearing (blue), AR-42-treated tumor-bearing (red), and combination-treated tumor-bearing (green) groups.

E   mRNA expression of WNT effectors upstream of β-catenin. Data are presented as mean ± SD of log-transformed fold-change (log2FC) values versus tumor-free controls. Groups: Tumor-free control (n = 6), tumor-free GTx-024-treated (n = 5), and tumor-bearing groups receiving vehicle (n = 4), GTx-024 (n = 5), AR-42 (n = 5), or combination (n = 6). *P < 0.1, **P < 0.05, ***P < 0.01, ****P < 0.001 based on Benjamini–Hochberg-adjusted P-values from DESeq2. Exact P-values provided in Appendix Table S8.

despite sufficient circulating levels of drug, demonstrated anabolic capability at the dose administered, and evidence of systemic hormonal activity (Figs EV1, EV2A, and EV4B). Our results align with the dearth of published reports of anabolic therapeutic efficacy in this common model. At the time of manuscript preparation, we identified a single demonstration of anti-cachectic anabolic therapy in C-26 mice (Morimoto et al, 2017), despite anabolic agents being among the most advanced clinical development programs in cancer wasting (Crawford et al, 2016; Argiles et al, 2017; Graf & Garcia, 2017).

Androgens have a well-characterized ability to normalize skeletal muscle catabolic gene expression associated with either glucocorticoid (dexamethasone)- or hypogonadism (castration)-induced atrophy (Jones *et al*, 2010; Serra *et al*, 2013; White *et al*, 2013a). We hypothesized that androgens' inability to reverse C-26 tumor-mediated atrogene expression underlies their lack of efficacy. Consistent with this hypothesis, inflammatory cytokine-driven catabolic signaling in male C-26 tumor-bearing mice appears completely insensitive to androgen administration. Another potential reason for exogenous androgens' failure is a cachexia-mediated global disruption of AR signaling. However, the response of the hypothalamic–pituitary–gonadal axis, several cytokines, and gastrocnemius transcriptome to androgen, along with no obvious effects of tumor burden on AR mRNA or protein (Fig 5B, Appendix Fig S5), suggests global AR blockade is unlikely. Alternatively in female C-26 tumor-bearing mice, single-agent GTx-024 showed trends in increased body weight and skeletal muscle mass, though not statistically significant (Fig 3). GTx-024 treatment in female mice had effects on pSTAT3 activation in skeletal muscle suggesting potential sex differences in anabolic resistance and IL-6/GP130/STAT3 axis sensitivity to androgens (Fig 6).

Lewis lung carcinoma tumor-bearing mice were also refractory to GTx-024 treatment. However, pSTAT3 activation in skeletal muscle was reduced in LLC as compared to C-26 tumor-bearing mice and treatment had no clear effect on pSTAT3 (Fig EV4). Cachectic drivers in the LLC and C26 models are thought to differ (Ballaro *et al*, 2016), and our results diverge from the reported benefits of ghrelin administration in this model (Chen *et al*, 2015). When taken together, these findings suggest drivers of anabolic resistance may be model-specific and exhibit treatment mechanism-dependent abilities to suppress anabolic signaling.

Androgens also have well-characterized direct anabolic effects on skeletal muscle that include targeting MUSCs and pluripotent mesenchymal progenitor cells to promote muscle hypertrophy (Dubois *et al*, 2012), suggesting compromised anabolic signaling might contribute to GTx-024's lack of anti-cachectic efficacy. All of our mechanistic analyses focused on gastrocnemius muscle, which readily responds to androgen administration despite scant AR expression (Serra *et al*, 2013). GTx-024 treatment in tumor-free mice resulted in very few DEGs, but subsequent GSEA (Subramanian *et al*, 2005), revealed robust β-catenin target gene regulation. Androgen-mediated β-catenin activation has been reported in whole muscle tissue (Gentile *et al*, 2010) and as a requirement for myogenic differentiation of pluripotent mesenchymal cells (Singh *et al*, 2009). Notably, GTx-024-mediated β-catenin target gene regulation was completely abrogated in the context of C-26-tumor burden, which corresponded to coordinated suppression of canonical WNT pathway effectors. GTx-024-mediated β-catenin activation was only restored in the presence of AR-42 which, as a monotherapy, normalized WNT effector expression (Fig 8D). Elucidating AR-42-responsive cachectic signals governing WNT suppression warrants further interrogation.

To the best of our knowledge, this is the first report of dysfunctional skeletal muscle WNT signaling in experimental cachexia. Importantly, both constitutive activation and genetic abrogation of WNT signaling impair proper adult MUSC function in response to injury (Otto *et al*, 2008; Rudolf *et al*, 2016; Agley *et al*, 2017). Our data suggest tightly controlled WNT signaling is lost in tumor-

bearing mice, which is consistent with other reports of MUSC dysfunction in the C-26 model (He *et al*, 2013). Given the clear effects of exogenous androgen administration on MUSC activation (Sinha-Hikim *et al*, 2003), it is plausible that disruption of WNT signaling represents a functional blockade of androgen-mediated anabolism in cachectic skeletal muscle (Fig 9). Furthermore, the dysfunctional WNT signaling reported here might be linked more broadly to the important clinical problem of cancer-induced anabolic resistance (Hardee *et al*, 2017).

We recognize that our experimental paradigm is limited in a number of ways. We evaluated only two models of cachexia and combination therapy were effective only in the C-26 model. The short treatment window (< 14 days) afforded by the C-26 model in our hands also severely curtailed our ability to demonstrate overt anabolism following GTx-024 treatment relative to other anabolic agents in less severe models of cachexia (Chen *et al*, 2015). We were additionally unable to monitor changes in animal body lean body mass over time. Though our endpoint measurements of skeletal muscle mass support our conclusion that improved anti-cachectic efficacy results from combined therapy, the ability to follow disease- and therapy-mediated changes in body composition longitudinally would provide improved context for our results. Lastly, we did not explore multiple dose levels in combination. Further dose optimization is essential to both improve efficacy and minimize toxicity as the tolerance for additional side effects ascribed to anti-cachexia therapy in heavily treated cancer patients is low.

### Combined anabolic and anti-catabolic therapy in cancer cachexia

To our knowledge, this is the first report combining SARM and HDAC inhibitor administration in experimental cachexia. In the C-26 model, we demonstrated efficacy using two agents currently undergoing clinical development, which included improved survival, total body weight, hindlimb skeletal muscle mass, and grip strength when SARM was combined with AR-42 over tumor-bearing controls and SARM monotherapy. Notably, in combination-treated female mice, terminal skeletal muscle weights were significantly improved compared to either single agent alone (Fig 3). Our mechanistic support for beneficial signaling in muscle following SARM and HDACi co-administration suggests that similar results are possible in male mice with optimized combination SARM regimens.

Despite established efficacy in diverse patient populations (Dalton *et al*, 2011; Dobs *et al*, 2013), GTx-024 failed to provide anabolic benefit in advanced NSCLC patients (Crawford, 2016). Though weight loss was not required for enrollment in GTx-024's registration trials, roughly half of all patients reported > 5% unexplained weight loss at initiation of chemotherapy suggesting a high prevalence of cachexia at diagnosis. In a similar cohort receiving anabolic ghrelin mimetic anamorelin therapy, subgroup analyses revealed patients with body mass indices < 18.5 (and presumably severe cachexia) showed no improvements in body composition (Temel *et al*, 2016). Analogous to these clinical populations, our data show that anabolic androgen administration cannot overcome severe catabolic signaling in the C-26 model and that profound cachectic burden additionally results in a blockade of critical anabolic signaling. Furthermore, we show that AR-42's anti-cachectic efficacy involves both mitigating catabolic signaling and

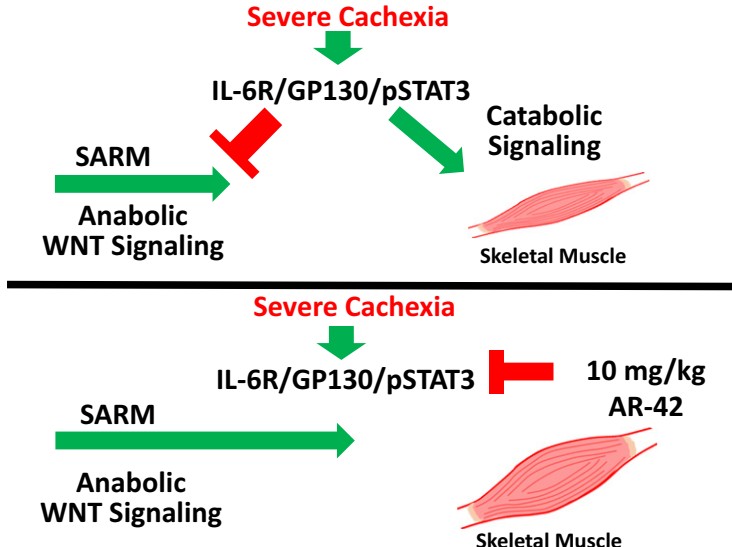

**Figure 9. Graphical mechanistic hypothesis.**

Cachexia-mediated IL-6 axis activation results in elevated catabolic signaling and disruption of WNT signaling in skeletal muscle. WNT dysfunction represents an apparent blockade of anabolic androgen signaling. Treatment with 10 mg/kg AR-42 modulates IL-6 axis activation and restores WNT responsiveness, promoting anabolism in skeletal muscle.

licensing anabolic signaling providing compelling mechanistic support for combined GTx-024/AR-42 administration in cachectic patients. Though catabolic drivers in the skeletal muscle of cachectic patients remain poorly characterized, IL-6 is thought to contribute to human cancer wasting (Baracos *et al*, 2018) and elevated circulating IL-6 has been shown to be increased in NSCLC patients and associated with reduced survival (Silva *et al*, 2017). These data support the relevance of AR-42's effects on the IL-6/GP130/STAT3 axis in cachectic patients and its use in combination with SARM therapy to improve anabolic response in patient populations with advanced cancer wasting.

## Materials and Methods

### Reagents and chemicals

GTx-024 [(S)-N-(4-cyano-3-(trifluoromethyl)phenyl)-3-(4-cyanophenoxy)-2-hydroxy-2-methylpropanamide] was synthesized as previously described (Kim *et al*, 2005), and its purity (99.26%) confirmed internally by LC-MS. AR-42 was generously provided by Arno Therapeutics, Inc. (Fairfield, NJ), and TFM-4-AS-1 (Sigma-Aldrich, Saint Louis, MO) and dihydrotestosterone (DHT; Steraloids, Newport, RI) were purchased from commercial sources.

Vehicle components included the following: Captex (Abitec, Columbus, OH), Tween-20 (Sigma-Aldrich, Saint Louis, MO), benzyl alcohol (Sigma-Aldrich, Saint Louis, MO), and sesame oil (Sigma-Aldrich, Saint Louis, MO). AR-42 was formulated in 0.5% methylcellulose [w/v] and 0.1% Tween-80 [v/v] in sterile water. GTx-024 (Narayanan *et al*, 2014), DHT, and TFM-4AS-1 (Schmidt *et al*, 2010) were formulated as previously described. Remaining reagents were all purchased from Sigma-Aldrich (Saint Louis, MO) unless otherwise mentioned.

### Cells

Cultured murine C-26 adenocarcinoma cells and LLC cells were maintained in fetal bovine serum (FBS)-supplemented (10%) RPMI 1640 medium or DMEM, respectively (Invitrogen, Carlsbad, CA), at 37°C in a humidified incubator with 5% $CO_2$. Cell lines were authenticated by using a commercial STR profiling service (Mouse Cell Authentication Service, ATCC 137-XV) and tested for mycoplasma by PCR using a commercial testing service (Mycoplasma PCR Testing Service, ATCC 136-XV). For injection into mice, cells were harvested by trypsinization, pelleted in FBS-supplemented culture medium, and then resuspended in sterile PBS at a concentration of $5 \times 10^6$ cells/ml. The C-26 and LLC cell lines were generously provided by the laboratory of Denis Guttridge (while at The Ohio State University) who had obtained the cells through a Material Transfer Agreement with the NCI (Bethesda, MD) in 2001. For all experiments, cells were cryopreserved at 7–10 passages and then passaged no more than four times after thawing from the same stock to propagate for injection into mice. This cell line has been maintained in the laboratories of co-authors Coss and Chen and has consistently produced cachexia, demonstrating that these cells have maintained their identity over time.

### Animals

Male CD2F1 mice (6–7 weeks of age), female CD2F1 mice (10–11 weeks of age), and male C57BL/6NHsd mice (7–8 weeks of age) (Envigo, Indianapolis, IN) were group-housed under conditions of constant photoperiod (12-h light/12-h dark), temperature, and humidity with *ad libitum* access to water and standard pelleted chow. Mice were briefly anesthetized (isoflurane) during administration of drugs (AR-42, GTx-024, vehicles) by oral gavage. In experiments in which food consumption was determined, the food in

each cage was weighed as frequently as once daily to once weekly and the decrease in food weight was divided by the number of mice in the cage. Tumor volumes were calculated from caliper measurements using a standard formula (length × width$^2$ × $\pi$/6). Mice were euthanized by $CO_2$ inhalation. All animal studies were conducted according to protocols approved by The Ohio State University Institutional Animal Care and Use Committee.

### Animal studies using the C-26 colon adenocarcinoma cachexia model

These studies were performed as previously described (Tseng et al, 2015a) with modifications. Tumors were established in the right flank of CD2F1 mice by subcutaneous injection of C-26 cells (0.5 × 10$^6$ cells in 0.1 ml). AR-42, GTx-024, and their vehicles were administered orally by gavage. TFM-4AS-1 (a potent experimental SARM), DHT, and vehicles were administered by subcutaneous injection.

#### AR-42 dose–response study
Emergent pharmacokinetic and tolerability data from humans receiving AR-42 (Sborov et al, 2017) suggested AR-42 exposures in human matching 50 mg/kg AR-42 exposures in mice would be poorly tolerated. Therefore the efficacy and associated exposures of reduced doses of AR-42 were determined. Male CD2F1 mice were stratified by body weight and then randomly assigned into five groups of six animals each. C-26 tumors were established in four of the groups, while those in the fifth group, serving as tumor-free controls, were injected with sterile saline. Six days later, animals with palpable tumors were treated with AR-42 once daily at 10 ($n = 5$) and 20 mg/kg ($n = 6$), and every other day at 50 mg/kg ($n = 5$, the originally described anti-cachectic dose; Tseng et al, 2015a), or vehicle control ($n = 4$) for 13 days. Upon sacrifice on study day 18, when the majority of tumor-bearing control mice met euthanasia criteria, the left gastrocnemius muscle was excised, flash-frozen in liquid nitrogen, and stored at −80°C for subsequent analyses. Carcass weights were corrected for tumor weight by assuming a tumor density equivalent to water (1 g/cm$^3$). Demonstration of anti-cachectic efficacy with the 10 mg/kg dose of AR-42 in this model prompted a follow-up experiment assessing the anti-cachectic efficacy of lower doses of AR-42. This study was performed as described above with animals treated with AR-42 once daily at 1, 3, and 10 mg/kg or with vehicle control ($n = 8$).

#### Initial AR-42/GTx-024 combination study (Study 1)
Male CD2F1 mice were stratified by body weight and then randomly assigned into six groups of six animals each. Historically, six animals per group provided sufficient power to detect treatment-mediated differences in tumor-bearing treated animals compared to controls. Tumors were established in four of the groups, while the fifth and sixth groups served as tumor-free controls. Six days later, animals with palpable tumors were treated twice daily for 13 days. AR-42 and its vehicle were administered in the mornings, and GTx-024 and its vehicle in the afternoons. Treatments included vehicles for AR-42 and GTx-024 ($n = 5$), GTx-024 (15 mg/kg; AR-42 vehicle; $n = 5$), AR-42 (10 mg/kg; GTx-024 vehicle; $n = 5$), or the AR-42 + GTx-024 combination (10 and 15 mg/kg, respectively; $n = 5$).

The remaining tumor-free groups received either vehicles ($n = 6$) or GTx-024 (15 mg/kg; AR-42 vehicle; $n = 6$). Body weight, tumor volume, and food consumption were monitored every other day. Upon sacrifice on day 18, sera were collected and hindlimb skeletal muscles, heart, spleen, and epididymal adipose tissues were harvested, weighed, flash-frozen, and stored for subsequent analyses.

#### Confirmatory AR-42/GTx-024 combination study (Study 2)
This confirmatory study was performed exactly as Study 1 with expanded animal numbers. Tumor-free control groups were maintained at 6 animals each, whereas 10 animals were included in each of the tumor-bearing groups. Six days after cell injection, animals with palpable tumors were treated as in Study 1 with vehicles ($n = 7$), GTx-024 ($n = 10$), AR-42 ($n = 9$), or the combination ($n = 9$). Grip strength was measured on study days 0 (baseline) and 16. Due to rapid model progression, this study was terminated on day 17 after only 12 days of treatment.

#### AR42/GTx-024 combination study in female mice (Study 3)
This study was performed exactly as Study 1 with female mice. The tumor-free control group contained six animals, whereas 12 animals were included in each of the tumor-bearing groups. Animals were treated as in Study 1 starting 6 days after cell injection with vehicle, GTx-024, AR-42, or the combination. Grip strength was measured on days 0 (baseline), 4, 11, and 18. Body weights were measured on days 0, 5, 11, 14, 16, and 18, and food consumption was monitored weekly. Similar to Study 1, sera, hindlimb skeletal muscles, spleen, and adipose were collected, weighed, and flash-frozen at sacrifice 18 days after cell injection.

#### Combined androgen and AR-42 study (Study 4)
Similar to Study 2, the tumor-free control group was maintained at six animals, whereas 10 animals were included in each of the 6 tumor-bearing groups. Six days after cell injection, animals with palpable tumors were treated once daily for 13 days with vehicles for AR-42 and TFM-4AS-1/DHT ($n = 9$), AR-42 (10 mg/kg; TFM-4AS-1/DHT vehicle; $n = 10$), TFM-4AS-1 (10 mg/kg; AR-42 vehicle; $n = 9$), DHT (3 mg/kg; AR-42 vehicle; $n = 10$), the combination of AR-42 and TFM-4AS-1 (10 mg/kg each; $n = 9$), or the combination of AR-42 (10 mg/kg) and DHT (3 mg/kg; $n = 10$). Grip strength was measured, and tissues were collected as in the previous studies.

### AR-42/GTx-024 combination study in the LLC cachexia model (Study 5)

This study was performed as previously described (Tseng et al, 2015a) with modifications. Tumors were established in the right hindlimb of male C57BL/6NHsd mice by intramuscular injection of LLC cells (0.5 × 10$^6$ cells in 0.05 ml). Similar to Study 3, the tumor-free control group was maintained at six animals, whereas 12 animals were included in each of the tumor-bearing groups. Six days after cell injection, animals were treated once daily with vehicle, AR-42 (10 mg/kg), GTx-024 (15 mg/kg), or the combination. Grip strength, body weight, and food consumption were measured, and tissues were collected as in the previous C-26 studies. This study was repeated as described with a lower dose of GTx-024 (0.5 mg/kg).

## Confirmation of anabolic activity of 15 mg/kg GTx-024

Eight-week-old, male, tumor-free CD2F1 mice were surgically castrated under isoflurane anesthesia. Three weeks later, the mice were assigned to groups treated with GTx-024 (15 mg/kg; $n = 12$) or vehicle ($n = 13$) by oral gavage once daily for 4 weeks. A sham-castrated group was treated identically with vehicle ($n = 14$). Body weights were measured twice weekly. Forelimb grip strength was measured 1 day prior to first treatment (baseline) and at end of study. At terminal sacrifice, hindlimb muscles were removed and weighed.

## Grip strength measurement

Forelimb grip strength was measured using Bio-GS3 Grip Strength Test Meter (Forceleader DBA Bioseb, Pinellas Park, FL). Each mouse was held by the base of its tail and lowered over the apparatus until its forepaws grasped the metal pull bar. The mouse was then gently pulled horizontally in a straight line away from the grip strength meter until the mouse released the bar. The force applied to the bar at the moment of release was recorded as the peak force. Five measurements were taken from each mouse, the average of which was designated as the mouse's grip strength.

## AR-42 plasma and tissue pharmacokinetics

Pharmacokinetic studies were performed as previously described (Cheng *et al*, 2016) with the following modifications. Seven week-old, male, tumor-free CD2F1 mice ($n = 3$ per dose and time point) were administered single oral doses of 10, 20, and 50 mg/kg AR-42 and then sacrificed 0.25, 4, and 24 h later. Fifty milligrams of gastrocnemius muscle tissue was flash-frozen in 2 ml BeadBlaster™ 24 (MIDSCI; St. Louis, MO) tubes and stored at −80°C. Samples were homogenized for 6 cycles using a BeadBlaster 24 with analytical standards in 1 mL methanol and then centrifuged at 15,600 $g$ (4°C) for 10 min. Supernatants were transferred to glass tubes, dried under nitrogen, and then reconstituted in 200 μl 40% methanol/0.1% formic acid. Plasma preparation and LC-MS/MS analyses were performed as previously described (Sborov *et al*, 2017). Mouse plasma and muscle pharmacokinetic parameters were estimated as previously described with the exception of $C_{avg}$, which was calculated as $AUC_{all}$/[dosing interval(h)].

## GTx-024 plasma pharmacokinetics

Male LLC tumor-bearing C57BL/6NHsd mice were administered a single oral 15 mg/kg dose of GTx-024 and then euthanized by $CO_2$ inhalation at 0.25, 0.5, 1, 2, 4, 8, 10, 24, and 48 h after dosing ($n = 2$ per time point, except $n = 1$ at 10 h). Blood was collected by cardiac puncture into heparinized tubes (BD Microtainer, Becton, Dickinson and Company, Franklin Lakes, NJ, USA) and immediately centrifuged at room temperature for 3 min at 4,200 RCF. Plasma was removed and placed on dry ice prior to storage at −80°C until analysis. Bioanalytical analyses were conducted as previously described (Cheng *et al*, 2016) by Charles River Laboratories, Inc. (Wilmington, MA, USA). Non-compartmental analysis was performed using the PKNCA package in R version 3.6.0.

## *In vitro* HDAC inhibition assays

HDAC activity was measured by a commercial vendor using human recombinant HDAC enzymes and fluorogenic HDAC substrates (Eurofins Cerep SA, Celle L'Evescault, France). Substrate concentrations ranged from 20 to 400 μM, and incubation conditions ranged from 10 to 90 min (RT or 37°C), depending on isoform. Results are expressed as percent inhibition of control specific activity in the presence of 1 μM AR-42.

## Cytokine analyses

Serum cytokine panel analyses were performed by a commercial vendor (Eve Technologies, Calgary, Canada) as previously described (Tseng *et al*, 2015a). Serum interleukin-6 (IL-6) was measured using a commercial ELISA kit (R&D Systems, Minneapolis, MN, USA) according to the manufacturer's instructions.

## Luteinizing hormone analyses

Luteinizing hormone was measured in serum by a two-site sandwich radioimmunoassay performed by the Ligand Assay and Analysis Core at the Center for Research in Reproduction, University of Virginia School of Medicine (Charlottesville, VA). Serum was isolated by centrifugation (2,000 $g$, 15 min) from whole blood samples collected from mice immediately post-mortem, and then stored at −80°C until shipment on dry ice.

## Western blot analyses

Western blots from all studies were performed on gastrocnemius muscle from representative animals lysed by Nonidet P-40 isotonic lysis buffer [50 mM Tris–HCl, pH 7.5, 120 mM NaCl, 1% (v/v) Nonidet P-40, 1 mM EDTA, 50 mM NaF, 40 mM glycerophosphate, and 1 g/ml each of protease inhibitors (aprotinin, pepstatin, and leupeptin)]. Equivalent amounts of protein from each sample, as determined by the Bradford assay (Bio-Rad), were resolved by SDS–PAGE and then transferred (semi-dry) onto Immobilon nitrocellulose membranes (Millipore, Bellerica, MA). Membranes were washed twice with TBST [Tris-buffered saline (TBS) containing 0.1% Tween-20], blocked with 5% non-fat milk in TBST for 1 h, and then washed an additional three times. Membranes were incubated with specific primary antibody in TBST (1:1,000) at 4°C overnight, washed three times (TBST), and then incubated with appropriate goat anti-rabbit or anti-mouse IgG–horseradish peroxidase-conjugated secondary antibodies (1:5,000) at room temperature (1 h). Following additional washes (TBST), immunoblots were visualized as appropriate by ECL chemiluminescence (Amersham Biosciences, Little Chalfont, United Kingdom) or Duration HRP Chemiluminescence (Alkali Scientific, Florida, United States). Primary antibodies: phospho-STAT3 (Tyr705) (D3A7) XP® rabbit mAb #9145, STAT3 (124H6) mouse mAb #9139 (Cell Signaling Technology, Danvers, MA); α-tubulin (B-7), sc-5286 (Santa Cruz Biotechnologies, Santa Cruz, CA); androgen receptor (EP670Y), ab52615 (Abcam, Cambridge, MA) and GAPDH (6C5), sc-32233 (Santa Cruz Biotechnologies, Santa Cruz, CA).

### Gene expression analyses—qRT–PCR

To generate muscle tissue RNA, 15 mg of gastrocnemius muscle tissue was lysed in 10 volumes of lysis buffer per tissue mass in prefilled 2.0-ml tubes with 3.0 mm zirconium beads (MIDSCI; St. Louis, MO). Tubes were loaded into BeadBlaster™ 24 (MIDSCI; St. Louis, MO) and centrifuged for five cycles of 5 s with a 30-s pause between cycles. Lysate was collected, and RNA was isolated using the mirVana™ miRNA Isolation Kit, with phenol (Thermo Fisher; Waltham, MA). Samples were treated with DNA-free DNA Removal Kit to eliminate any DNA contamination (Invitrogen, Carlsbad, CA). Total RNA (0.5 µg) was reverse-transcribed using high-capacity cDNA reverse transcription kit (Applied Biosystems, Foster City, CA) for 10 min at 25°C, 120 min at 37°C, and 5 min at 85°C (T100™ Thermal Cycler, Bio-Rad). Real-time qPCR was performed on the QuantStudio 7 system (Applied Biosystems, Foster City, CA) using the powerup SYBR Green Master Mix (Applied Biosystems, Foster City, CA). Cycling was performed using the QuantStudio 7 real-time PCR software—2 min at 50°C and 10 min at 95°C—followed by 40 cycles of 15 s at 95°C and 1 min at 60°C. All real-time qPCR assays were carried out using technical duplicates using β-actin or GAPDH as the internal control genes. Reaction specificity was supported by the detection of a single amplified product in all reactions by a post-cycling melt curve, the absence of non-template control signal for 40 cycles, and confirmation of amplicon size using agarose electrophoresis. Primers for analyses are listed in Appendix Table S14. Data are presented as per group geometric mean ± geometric standard deviation (STD) and individual $2^{-\Delta C_t}$ values [$\Delta C_t$ = (target gene − internal control)]. All values are normalized to the geometric mean tumor-free control values. As transformed expression data are not normally distributed, statistical differences between treatment groups were determined by one-way ANOVA followed by Dunnett's test on raw delta $C_t$ values.

### Pathway enrichment analyses

Overlap between the gene sets in Tseng et al (2015a) (Data ref: Tseng et al, 2015b) and Bonetto et al (2011) (Data ref: Zimmers et al, 2011) was determined and plotted using the Venn diagram web tool (http://bioinformatics.psb.ugent.be/webtools/Venn/) in the Bioinformatics and Evolutionary Genomics Suite (Ghent University, Ghent, Belgium). The lists of AR-42-regulated genes from Tseng et al, as well as genes regulated in common between Tseng et al and Bonetto et al, were tested for significant overlap with canonical pathway gene sets using the 'Compute Overlaps' function from the Molecular Signatures Database (Broad Institute, Cambridge, MA; software.broadinstitute.org/gsea/msigdb/annotate.jsp). Redundant results were collapsed to show only the gene set with the largest number of genes.

### Gene expression analyses—RNA-seq

For each sample, 600 ng total RNA (bioanalyzer RIN values > 7) was used to generate polyA-enriched RNA-seq libraries with the NEBNext Ultra II Directional RNA Library Prep Kit (New England Biolabs, Inc., catalog number E7760) and sequenced as PE-150 reads on HiSeq 4000 (Illumina Inc, San Diego, CA). Raw fastq files

were adaptor-trimmed using Trimmomatic (Bolger et al, 2014), aligned to the mm10 genome with Subread (Liao et al, 2013), and marked for duplicate reads with Picard v2.3.0 (http://broadinstitute.github.io/picard/). Samtools (Li et al, 2009) was used to calculate post-alignment quality control metrics (Appendix Table S15). A gene-based counts matrix was generated with the summarizeOverlaps function of GenomicAlignments (Lawrence et al, 2013) and analyzed with DESeq2 (Love et al, 2014) for genes differentially expressed among groups. GSEAPreranked analyses (Subramanian et al, 2005) were performed using the signed log10 P-values for each gene to test for significant enrichment of gene sets in the 'TF Targets' file downloaded from ge-lab.org/#/data. Any duplication in signed log10 P-values was removed prior to running the GSEAPreranked analyses by adding a small value sampled at random from a normal distribution with mean zero and standard deviation 0.00001 to each duplicated value. GSEAPreranked analyses were performed in classic mode on gene sets with a minimum and maximum of 5 and 5000 genes in the gene set, respectively, and were followed by leading edge analyses for selected comparison and gene set combinations. Heatmaps for RNA-seq data were generated with pheatmap (Kolde, 2015) using z-scores calculated from log count-per-million values obtained with the 'cpm' function in edgeR (Robinson et al, 2010).

### Phospho-STAT3/STAT3 analysis

Expression of phospho-STAT3 and total STAT3 in C-26 tumor tissues was measured using commercial sandwich ELISA kits according to the manufacturer's instructions (PathScan® Phospho-Stat3, #7300, and Total Stat3, #7305, Cell Signaling Technology, Danvers, MA).

### Cell viability assay

Cell viability assays were conducted using the CCK-8 kit (Dojindo Molecular Technologies, Inc; Kumamoto, Japan). Briefly, 1,250 cells per well were plated on 96-well plates and allowed to adhere overnight. Cells were treated with DHT or GTx-024 (0.001–10 µM) for 24 or 48 h and then assayed according to the manufacturer's protocol.

### Statistical methodology

Plotting and statistical analyses were performed using GraphPad Prism version 7 (GraphPad Software, La Jolla, CA). The specific statistical tests employed are outlined in detail within the figure legends. Briefly, data were analyzed with a one-way ANOVA, followed by multiplicity adjusted P-values using either Tukey's or Dunnett's methodology (Wright, 1992) as noted in the figure legends. $P < 0.05$ were considered significant, and significantly different pairwise comparisons were annotated as noted in the figure legends. Survival data were analyzed by the log-rank (Mantel–Cox) test, and multiple comparisons were performed using a Bonferroni-corrected threshold. $P < 0.016$ were significant for multiple comparisons.

## Data availability

The RNA-seq data presented in this article have been deposited in the NCBI Gene Expression Omnibus (Wright, 1992) and are accessible

### The paper explained

#### Problem
Cancer cachexia is a multifactorial syndrome characterized by involuntary muscle loss leading to decreased tolerance of chemotherapy, lowered quality of life, and reduced overall survival. Given the complex pathophysiology of this syndrome, there is an unmet need to develop multimodal intervention strategies in advanced cancer patients.

#### Results
The present study demonstrates: (i) consistent with clinical experience in NSCLC patients, anabolic therapy GTx-024 is ineffective as a monotherapy in murine models of cancer wasting, (ii) combined treatment with AR-42, a clinically evaluated HDAC inhibitor, and GTx-024 demonstrated improved anti-cachectic efficacy, (iii) the anti-cachectic efficacy of AR-42 is associated with limiting IL-6 axis activation, (iv) GTx-024-mediated β-catenin target gene regulation in skeletal muscle is abrogated in the presence of cachectic tumor burden, and (v) permissive anabolic WNT signaling in experimental cachexia is restored following combination treatment of AR-42 and GTx-024.

#### Impact
Our data suggest anabolic signaling as a result of androgen administration cannot overcome the catabolic signaling associated with cachectic burden in the C-26 murine model of cancer wasting. Furthermore, AR-42 demonstrates anti-cachectic efficacy through alleviating IL-6 axis-mediated catabolic signaling and restoring anabolic sensitivity in cachectic skeletal muscle. These data provide support for combined AR-42/GTx-024 administration as a potential therapeutic avenue to combat the complex mechanisms underlying wasting in advanced cancer patients.

through GEO Series accession number GSE138464 (https://www.ncbi.nlm.nih.gov/geo/query/acc.cgi?acc=GSE138464).

**Expanded View** for this article is available online.

## Acknowledgements
We are grateful to Dr. Appaso Jadhav and Ms. Uma Subrayan (The Ohio State University College of Pharmacy) for verification of GTx-024 purity and technical assistance in preparation of tissue samples for analysis, respectively. We thank Arno Therapeutics, Inc., for generously providing AR-42. We also thank the Molecular Carcinogenesis and Chemoprevention Program (OSU Comprehensive Cancer Center), as well as Dr. Jiang Wang and the Pharmacoanalytical Shared Resource and Genomics Shared Resource in The Ohio State University Comprehensive Cancer Center, which is supported by NCI/NIH Grant P30-CA016058. The luteinizing hormone assays were performed by the University of Virginia Center for Research in Reproduction Ligand Assay and Analysis Core, which is supported by the Eunice Kennedy Shriver NICHD/NIH (NCTRI) Grant P50-HD28934. This work was also supported in part by NCI/NIH K12-CA133250-07 (Coss), Eli Lilly Fellowship (Liva), and Pelotonia Idea Award (OSU Comprehensive Cancer Center, Coss).

## Author contributions
Conceived the study: CCC, C-SC; Designed the experiments: CCC, C-SC; Performed experiments and acquired data: SGL, Y-CT, AMD, SKK, BCR, TV, SEH, Y-CK; Data analysis: MGS, JAB, XZ, MC, MAP, CCC; Wrote, edited the manuscript: CCC, C-SC, SGL, SKK, Y-CT, TV, MAP, TB-S.

## Conflict of interest
C.-S. Chen is an inventor of AR-42, which was licensed to Recursion Pharmaceuticals, Inc., for clinical development by The Ohio State University Research Foundation. C.C. Coss is a former employee of GTx, Inc., owner of GTx-024, but has no financial relationship with or any equity in GTx, Inc. Recursion Pharmaceuticals, Inc. and GTx, Inc. were not involved in any way with the financing, design, or interpretation of the reported studies. All other authors declare that they have no conflict of interest.

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
