## [Review Process File · EMBO Molecular Medicine]

Overcoming Resistance to Anabolic SARM Therapy in Experimental Cancer Cachexia with an HDAC Inhibitor

Sophia G. Liva, Yu-Chou Tseng, Anees M. Dauki, Michael G. Sovic, Trang Vu, Sally E. Henderson, Yi-Chiu Kuo, Jason A. Benedict, Xiaoli Zhang, Bryan C. Remaily, Samuel K. Kulp, Moray Campbell, Tanios Bekaii-Saab, Mitchell A. Phelps, Ching-Shih Chen and Christopher C. Coss

Review timeline:

Submission date:	14 October 2018
Editorial Decision:	8 November 2018
Revision received:	16 August 2019
Editorial Decision:	13 September 2019
Revision received:	26 November 2019
Accepted:	29 November 2019

Editor: Lise Roth

Transaction Report:

1st Editorial Decision

8 November 2018

Thank you for the submission of your manuscript to EMBO Molecular Medicine. We have now heard back from the two referees whom we asked to evaluate your manuscript.

As you will see from the reports below, both referees mention the potential high medical impact and interest of the study, but they also have crucial overlapping concerns that should be addressed in a major round of revision of the present manuscript, so that the data fully support the conclusions. In particular, other model(s) of cancer cachexia should be included, and data presentation (including adequate statistical tests) should be thoroughly improved.

Addressing the reviewers concerns in full will be necessary for further considering the manuscript in our journal. EMBO Molecular Medicine encourages a single round of revision only and therefore, acceptance or rejection of the manuscript will depend on the completeness of your responses included in the next, final version of the manuscript.

EMBO Molecular Medicine has a "scooping protection" policy, whereby similar findings that are published by others during review or revision are not a criterion for rejection. Should you decide to submit a revised version, I do ask that you get in touch after three months if you have not completed it, to update us on the status. Please also contact us as soon as possible if similar work is published elsewhere. If other work is published, we may not be able to extend the revision period beyond three months.

I look forward to receiving your revised manuscript.

***** Reviewer's comments *****

Referee #1 (Comments on Novelty/Model System for Author):

To improve the model I would add data , using the drugs in an animal model more resembling that where the drug was used in NSCLC in humans. Please refer to Proc Natl Acad Sci U S A. 2018 Jan 23;115(4):E743-E752. doi: 10.1073/pnas.1714703115. Epub 2018 Jan 8.

Fenofibrate prevents skeletal muscle loss in mice with lung cancer.

Goncalves MD1,2,3, Hwang SK1,2, Pauli C4, Murphy CJ1,2,5, Cheng Z6, Hopkins BD1,2, Wu D1,2, Loughran RM1,2, Emerling BM1,2, Zhang G6, Fearon DT1,2,7, Cantley LC8,2.

Referee #1 (Remarks for Author):

The authors investigate the mechanism behind failure of androgen-based therapy in cancer cachexia. They present a deep effort in understanding the mechanism and have solid and robust data for the combined therapy that shall be recommended to cancer patients at risk of developing cachexia. The "Discussion" shall be enriched as proposed below and some experiments shall be done before publication of this very important work for the medical community occurs.

Minor comments:

- Please show the piece of data relating to these lines:

...in our hands, SARMs displayed essentially no impact on muscle wasting associated with the common colon-26 (C-26) mouse model of experimental cancer cachexia (unpublished).

- Give an explanation about how study 2 results in bigger tumors and premature euthanasia than study 1

(different n of cells injected? Selection of a more aggressive clone of C26 after culturing in vitro the cells, which passage were these cells? How many times at maximum were C26 cells trypsinized before injecting in vivo? These details shall be added in Mat and Met section)

- Is IL6 increased in plasma of non small cell lung cancer patients? This would support a link between their studies and clinical impact of the combined treatment they propose (this shall be added in the Discussion)

- In the discussion, there is an overstatement that is not supported by adequate citations ..despite anabolic agents representing the most advanced clinical development programs in cancer wasting [35].

Please change this line using more cautious tone and add references as well

Major comments:

- They show data in males, what about C26-carrying female mice treated in the same ways?

- Why DHT alone resulted in smaller C26 tumor? Please try to give an explanation. Is the androgen receptor expressed by C26 tumor? To my knowledge the C26 comes from a female BalB C mouse (NCI), did they check the sex of the C26 cells? If a male counterpart exists, do they obtain similar results with the treatments of mice carrying this male-derived C26?

- To improve the model I would add data (at least for the combined treatment), using the proposed drugs in an animal model more resembling that where the drug was used in NSCLC in humans and failed. Please refer to Proc Natl Acad Sci U S A. 2018 Jan 23;115(4):E743-E752. doi: 10.1073/pnas.1714703115. Epub 2018 Jan 8.

Fenofibrate prevents skeletal muscle loss in mice with lung cancer.

Goncalves MD1,2,3, Hwang SK1,2, Pauli C4, Murphy CJ1,2,5, Cheng Z6, Hopkins BD1,2, Wu D1,2, Loughran RM1,2, Emerling BM1,2, Zhang G6, Fearon DT1,2,7, Cantley LC8,2.

- Survival plots with single and combined treatments shall be provided for the C26 model they employed

Referee #2 (Comments on Novelty/Model System for Author):

needs statistical review

Needs clarity and focus

only 1 preclinical clinical model decreases relevance broadly

Referee #2 (Remarks for Author):

The working hypothesis is the combining SARM/AR-32 therapy will suppress cachexia in the preclinical C-26 models of cancer cachexia. Rationale is based on the minimal effect of the SARM in the mice administered C-26 cells. However, HDAC inhibitor AR-42 has demonstrated benefits. The authors report that AR-42 suppresses catabolic gene expression and improves SARM responsiveness

Specific

This work extends a prior investigation examining the anti-cachexia properties of HDACi in the C-26 cancer cachexia model.

Overall the statistics used, and design of the figures make the data very difficult to interpret. The authors should strive for more clarity in the presentation of their results and also emphasize the critical points directly related to the overall purpose. The results would benefit from more sophisticated measurements of lean body mass over time. Related to muscle mass of the quad and gastrocnemius, AR-42+GTx-024 does not appear different from either alone, which directly impacts the interpretation. It appears the very large intra-animal variability related to all circulating variables make interpretations about treatment drug treatment effects unmanageable please change the results to reflect this reality.

There are several key points that this manuscript identifies in the discussion that could better demonstrated to the reader if the overall presentation was more focused including the comparisons and design of the figures. There is also an over reliance on supplemental data that speaks to the broad presentation. The authors are encouraged to focus the study on a few key measurable outcomes that are impactful and then carefully present them to the reader. The authors need to work to improve the rigor of the analysis.

Introduction, page 5, remove statements involving unpublished data related to C-26

The study rationale should be based off of published data and needs strengthened in the introduction. Please rework.

The introduction hypothesis is not a hypothesis which is predictive and directional showing relationship between key variables. Please edit or restate as an objective.

The introduction's broad importance is cachexia, but the premise is delimited to the C-26 model. Please provide context for the C-26 model in relation to other preclinical models. Would your results with this model have implications for other models?

Methods. Provide a reference and rationale for the AR-42 dosing paradigm and relevance of the doses in the dose response study.

Statistical methodology. Please supply more information in this section. Please clarify that ANOVA 2 way was performed (tumor and drug treatment). Please provide rationale for the multiple comparison tests used.

Page 9, please clarify that the dose "reversed", or did it block or prevent? To reverse means you have data that there was a significant drop that was then induced. Please plot body weight change over time. Please add lean body mass measurements over time related to DEXA or similar methodology.

Results, Figure 2. Please include data for AR-42 with no tumor. Please provide the N for the table data.

Please clarify the relationship between figure 4D and Figure 4F, the graph appears more related to changes in total stat and the control is overloaded and not useful. Were the changes in total STAT expected?

Figure 4 A is a table.

1st Revision - authors' response

16 August 2019

***** Reviewer's comments *****

Referee #1 (Comments on Novelty/Model System for Author):

To improve the model I would add data , using the drugs in an animal model more resembling that where the drug was used in NSCLC in humans. Please refer to Proc Natl Acad Sci U S A. 2018 Jan 23;115(4):E743-E752. doi: 10.1073/pnas.1714703115. Epub 2018 Jan 8.

Fenofibrate prevents skeletal muscle loss in mice with lung cancer.

Goncalves MD1,2,3, Hwang SK1,2, Pauli C4, Murphy CJ1,2,5, Cheng Z6, Hopkins BD1,2, Wu D1,2, Loughran RM1,2, Emerling BM1,2, Zhang G6, Fearon DT1,2,7, Cantley LC8,2.

Referee #1 (Remarks for Author):

The authors investigate the mechanism behind failure of androgen-based therapy in cancer cachexia. They present a deep effort in understanding the mechanism and have solid and robust data for the combined therapy that shall be recommended to cancer patients at risk of developing cachexia. The "Discussion" shall be enriched as proposed below and some experiments shall be done before publication of this very important work for the medical community occurs.

Minor comments:

Reviewer Comment: Please show the piece of data relating to these lines:

...in our hands, SARMs displayed essentially no impact on muscle wasting associated with the common colon-26 (C-26) mouse model of experimental cancer cachexia (unpublished).

Author Response: We appreciate this comment from the reviewers. As one of our key findings involves this phenomenon, a thorough analysis is presented within the paper. We have edited this sentence and removed references to unpublished data from the current version of the manuscript.

Reviewer Comment: Give an explanation about how study 2 results in bigger tumors and premature euthanasia than study 1

(different n of cells injected? Selection of a more aggressive clone of C26 after culturing in vitro the cells, which passage were these cells? How many times at maximum were C26 cells trypsinized before injecting in vivo? These details shall be added in Mat and Met section)

Author Response: As suggested by the reviewer, we have included more details concerning the culture of C-26 cells prior to injection in the Supplementary Materials and Methods section (see Cells subsection) including the range of C-26 cell passages used in these studies. The number of cells injected was the same in each experiment.

As others have shown, C-26 cells' cachectogenic phenotype can vary based on small changes in cell handling (MethodsX. 2015;2:53-58). In our hands, the C-26 model performs similarly to others at our institution with mean tumor volumes ranging between ~0.75 to ~1.0 grams at 2-3 weeks post-injection (J Cachexia Sarcopenia Muscle. 2014 Dec;5(4):321-8; Am J Physiol Heart Circ Physiol. 2017 Jun 1;312(6):H1154-H1162). Given no obvious change in how the C-26 cells were handled between the experiments presented, we hypothesize that perhaps involvement of different lab members contributed to between-experiment variability that resulted in the 1-day difference in euthanasia time point.

Reviewer Comment: Is IL6 increased in plasma of non small cell lung cancer patients? This would support a link between their studies and clinical impact of the combined treatment they propose (this shall be added in the Discussion)

Author Response: The authors appreciate this suggestion to improve the Discussion by adding support for the clinical relevance of IL-6 in patients with cachexia-related malignancy. Circulating IL-6 has been shown to be increased in NSCLC patients and associated with reduced survival (PLoS One. 2017 Jul 17;12(7):e0181125). IL-6 is also thought to contribute to cancer cachexia in humans (Nat Rev Dis Primers. 2018 Jan 18;4:17105). This important additional information has been included in the updated Discussion (see the last paragraph of the Discussion, p27).

Reviewer Comment: In the discussion, there is an overstatement that is not supported by adequate citations

...despite anabolic agents representing the most advanced clinical development programs in cancer wasting [35].

Please change this line using more cautious tone and add references as well

Author Response: The authors agree that this statement can be vastly improved with a more cautious tone and improved references. In the revised manuscript, we have edited the sentence to the following, including the removal of reference #35 and addition of new references: "...despite anabolic agents **being among** the most advanced clinical development programs in cancer wasting (Argiles et al, 2017; Crawford et al, 2016; Graf & Garcia, 2017)." (see Discussion, Impact of cachectic tumor burden on androgen signaling, last sentence of the first paragraph, p22)

Major comments:

Reviewer Question: They show data in males, what about C26-carrying female mice treated in the same ways?

Author Response: The authors appreciate this question concerning the sex specificity of both resistance to androgen therapy and response to our therapeutic approach. We repeated our entire experimental paradigm in female mice bearing C-26 tumors, the data from which are **presented in Figure 3 and 6D** in the updated manuscript. In short, female mice also exhibited anabolic resistance, but an improved response to combination therapy relative to male mice. Relevant information has been added to the Materials and Methods (p29, *AR-42/GTx-024 Combination Study in Female Mice (Study 3)*), the Supplementary Materials and Methods (p3, *Animals* section), the Results (p10, paragraph starting with, "Male mice are generally..."; and p16, 3rd line from top, "Female mice...") and throughout the Discussion sections to highlight these findings and discuss them in the context of our other results (*Discussion*, p23, p26).

Reviewer Question: Is the androgen receptor expressed by C26 tumor?

Author Response: We performed an androgen receptor (AR) western blot on both types of tumor cells used in the updated manuscript (C-26 and LLC). We evaluated lysates from both cultured cells and tumors resulting from their injection into mice and found that C-26 cells and tumors are AR-positive, whereas LLC cells and tumors are AR-negative. These results are **presented in Supplementary Figure 7B** in the updated manuscript and we added description/discussion of this data in the context of the observation that tumors were smaller in DHT-treated mice (see *Results*, p12, first paragraph, passage starting with, "We determined that..." Additional discussion is presented in the following Author Response).

Reviewer Question: Why DHT alone resulted in smaller C26 tumor? Please try to give an explanation.

Author Response: The authors appreciate the need to explain this unexpected result. After determining that C-26 cells expressed the androgen receptor, we performed cellular proliferation assays in the presence of both GTx-024 and DHT. We did not detect differences in the direct effects of these androgens on C-26 cell growth that would explain the smaller tumor volumes in DHT-treated mice. These data are **presented in Supplementary Figure 7C** in the updated manuscript. These data suggest that DHT's effects in C-26 tumor growth are indirect. There is some evidence that DHT is differentiated from testosterone in its effects on circulating immune cells (J Appl Physiol (1985). 2003 Jul;95(1):104-12). It is tempting to speculate that systemic DHT administration may alter how the mouse's immune system responds to C-26 tumors in ways SARM

administration does not. Alternatively, given DHT's increased potency, perhaps higher doses of SARM would similarly suppress tumor burden as monotherapy.

Reviewer Question: To my knowledge the C26 comes from a female BalB C mouse (NCI), did they check the sex of the C26 cells? If a male counterpart exists, do they obtain similar results with the treatments of mice carrying this male-derived C26.

Author Response: The reviewer is correct; C-26 cells are indeed female in origin (Cancer Res. 1975 Sep;35(9):2434-9) and, to the best of our knowledge, no male-derived C-26 counterpart exists.

Reviewer Question: To improve the model I would add data (at least for the combined treatment), using the proposed drugs in an animal model more resembling that where the drug was used in NSCLC in humans and failed. Please refer to Proc Natl Acad Sci U S A. 2018 Jan 23;115(4):E743-E752. doi: 10.1073/pnas.1714703115. Epub 2018 Jan 8.

Fenofibrate prevents skeletal muscle loss in mice with lung cancer.

Goncalves MD1,2,3, Hwang SK1,2, Pauli C4, Murphy CJ1,2,5, Cheng Z6, Hopkins BD1,2, Wu D1,2, Loughran RM1,2, Emerling BM1,2, Zhang G6, Fearon DT1,2,7, Cantley LC8,2.

Author Response: The authors appreciate this comment and through the editor obtained approval from the reviewer(s) to utilize the well characterized Lewis Lung Carcinoma (LLC) model of cancer cachexia (Semin Cell Dev Biol. 2016 Jun;54:20-7) instead of the suggested genetic model of lung cancer. The results of these additional studies are presented in the Supplementary Figure S6 in the updated manuscript. Despite LLC cells being AR-negative, we found LLC tumor growth to be stimulated by 15 mpk GTx-024 administration (Figure S6A) which confounded our ability to evaluate anti-cachectic effects and required a dose reduction of GTx-024. We repeated the experiment using GTx-024 at 0.5 mpk which is expected to be fully anabolic based on the previous extensive work on GTx-024's anabolic pharmacology in rats (J Pharmacol Exp Ther. 2005 Oct;315(1):230-9) and the mouse pharmacokinetic GTx-024 data we generated for this resubmission (Figure S6B). These new pharmacokinetic data show that an oral 15 mpk dose of GTx-024 in mouse results in an AUCinf of 863 ug*h/ml suggesting that a 0.5 mpk dose would generate an AUC of 28.8 ug*hr/mL (30-fold reduction). Kim et al (J Pharmacol Exp Ther. 2005 Oct;315(1):230-9.) have shown that 0.5 mpk doses of GTx-024 are fully anabolic in male rats and are associated with an AUCinf of approximately 4.3 ug*hr/mL (Xenobiotica. 2013 Nov;43(11):993-1009). Therefore, it is reasonable to expect that a 0.5 mpk oral dose in mice, expected to generate 6.7-fold higher exposures should similarly be fully anabolic. However, as noted, at this lower dose, LLC tumor-bearing mice appear resistant to anabolic androgen therapy alone or in combination with AR-42 (Figure S6D,E). These results have been added to the updated manuscript and discussed in the context of our other findings (see Supplementary Figure S6; Results, Anabolic Resistance in the LLC Model, p10-11; and Discussion, Impact of cachectic tumor burden on androgen signaling, 2nd paragraph, p22 ["In the LLC model,..."], and p23, last paragraph ["As mentioned, GTx-024 treatment in male LLC tumor-bearing..."]).

Though our combination approach was not effective at the doses administered in LLC tumor-bearing animals, both commonly employed cachexia models (C-26 and LLC) appear resistant to the anabolic effects of androgen administration on skeletal muscle. Cachexia in the LLC model is thought to be driven by TNF alpha (Mol Cell Endocrinol. 1998 Jul 25;142(1-2):183-9.), which suggests there may be multiple drivers of anabolic resistance. As with cancer therapy, it is likely that specific anti-cachectic therapies will only be effective in subsets of cachectic patients. Our data suggest that wasting associated with strong IL-6 axis activation would be suitable for combined SARM/AR-42 treatment.

Reviewer Question: Survival plots with single and combined treatments shall be provided for the C26 model they employed

Author Response: As a matter of practice, due to rapid tumor growth in our models, all of our mice are sacrificed at the same time. However, utilizing $\geq 20\%$ body weight loss (a common euthanasia cut-off) as surrogate survival criteria, we can evaluate survival as we have done in a previous report (J Natl Cancer Inst. 2015 Oct 12;107(12):djv274). This analysis demonstrated improved survival for GTx-024/AR-42 combination-treated C-26 tumor-bearing mice. These results are presented in Figure 2E in our revised manuscript.

Referee #2 (Comments on Novelty/Model System for Author):

needs statistical review

Needs clarity and focus

only 1 preclinical clinical model decreases relevance broadly

Referee #2 (Remarks for Author):

The working hypothesis is the combining SARM/AR-32 therapy will suppress cachexia in the preclinical C-26 models of cancer cachexia. Rationale is based on the minimal effect of the SARM in the mice administered C-26 cells. However, HDAC inhibitor AR-42 has demonstrated benefits. The authors report that AR-42 suppresses catabolic gene expression and improves SARM responsiveness

Specific

This work extends a prior investigation examining the anti-cachexia properties of HDACi in the C-26 cancer cachexia model.

Reviewer Comment: Overall the statistics used, and design of the figures make the data very difficult to interpret. The authors should strive for more clarity in the presentation of their results and also emphasize the critical points directly related to the overall purpose.

Author Response: The authors agree that our analysis and findings could be presented more clearly. To this end, we made several key changes in data presentation.

- 1) Data from the various animal studies were gathered together into individual figures and data were moved from the supplements to the main manuscript such that the results of each animal study could be more readily interpreted. We also reduced the number of tissues we presented.
- 2) Animal body weights, tissue weights and functional assessments are all presented as % of Tumor Free controls in the updated manuscript (Figures 1F-H; Figures 2A-D, Figures 3A-D, Figures 4A-D), where they had previously been presented as a mix of comparisons to baseline and to % of Tumor Free controls.
- 3) Statistical analysis was harmonized across multiple studies as were notations of statistical differences. Specifically, animal data were analyzed with a one way ANOVA followed by between group pair-wise comparisons with Tukey's correction for multiple comparisons or Dunnett's multiple comparison test. Despite all pair-wise group comparisons being made, for clarity in presentation, only differences ($p < 0.05$) between group means and the tumor-bearing vehicle-treated controls (V), tumor-bearing GTx-024-treated (G) mice, and tumor-bearing AR-42-treated (A) mice are noted on the plots.

Please note that these extensive changes are integrated throughout the manuscript making it difficult in this letter to point the reviewers and editors to the specific locations of these changes. All of these changes are tracked in the revised manuscript.

Reviewer Comment: The results would benefit from more sophisticated measurements of lean body mass over time.

Author Response: The authors agree that longitudinal assessment of lean body mass would improve the manuscript. Regrettably, we do not have access to instrumentation that readily assesses lean mass in the models we present. Along with others at our institution (Oncol Rep. 2019 May;41(5):2909-2918), we have determined that our small animal EchoMR cannot readily detect lean mass deficits in C-26 tumor-bearing mice.

Reviewer Comment: Related to muscle mass of the quad and gastrocnemius, AR-42+GTx-024 does not appear different from either alone, which directly impacts the interpretation.

Author Response: In the updated manuscript, combination-treated mice demonstrated statistically significantly larger gastrocnemius muscle than mice receiving GTx-024 monotherapy (Figure 1G, Figure 2B, and Figure 3B) or AR-42 monotherapy (Figure 3B). Similarly, combination-treated mice demonstrated statistically significantly larger quadriceps muscle than GTx-024 monotherapy (Figure

2C, Figure 3C) or AR-42 monotherapy (Figure 3C). Importantly, in tumor-bearing female mice, combination therapy improved both gastrocnemius and quadriceps mass relative to both GTX-024 and AR-42 monotherapies (Figure 3).

Reviewer Comment: It appears the very large intra-animal variability related to all circulating variables make interpretations about treatment drug treatment effects unmanageable please change the results to reflect this reality.

Author Response: The authors agree that the variability in circulating cytokines etc. is large and have updated the Results section to explicitly point out this variability and its potential impact on data interpretation (see *Results*, p14, last line, “Although inter-animal variability...”).

We were unable to assess intra-animal variability as each assessment comes from a terminal blood draw at the end of a study, but we appreciate that the inter-animal variability is quite large. Despite this large variability, using standard statistical approaches (one way ANOVA, Dunnett’s correction for multiple comparisons), treatment with GTX-024 in tumor-bearing animals resulted in statistically significant increases in circulating LIF and decreases in G-CSF and IP-10 (Table 1 in the updated manuscript). Similar coefficients of variation existed for AR-42-treated animals for these cytokines as compared to GTX-024-treated animals (i.e. ~78% versus 68%, respectively, for G-CSF, ~32% versus %31, respectively, for LIF, etc.), but significant AR-42-mediated differences in these parameters were not detected.

Reviewer Comment: There are several key points that this manuscript identifies in the discussion that could better demonstrated to the reader if the overall presentation was more focused including the comparisons and design of the figures.

Author Response: As noted above, in an attempt to streamline the presentation of the data and to make the results more readily accessible, the statistical tests used and the presentation of results were harmonized across the multiple animal studies presented. The discussion in the updated manuscript was similarly streamlined to highlight and focus on key findings.

Reviewer Comment: There is also an over reliance on supplemental data that speaks to the broad presentation.

Author Response: In the updated manuscript, the majority of Supplementary Figure 4 was moved into the main manuscript. Moreover, if we are fortunate enough to have this paper accepted, we will take advantage of the Expanded View feature of this journal so that figures of greater value that are currently included among the supplementary data will be more readily accessible to readers.

Reviewer Comment: The authors are encouraged to focus the study on a few key measurable outcomes that are impactful and then carefully present them to the reader.

Author Response: In the updated manuscript, we have made an effort to highlight the key findings, dedicating a separate subsection in the Discussion to each, which include increased insight into the anti-cachectic mechanism of AR-42 (p20), the resistance of cachectic mice to anabolic androgen administration (p22), and the ability of co-administration of SARM with an HDAC inhibitor to improve anti-cachectic efficacy (p26).

Reviewer Comment: The authors need to work to improve the rigor of the analysis.

Author Response: We included an additional experienced biostatistician (Xiaoli Zhang) as a co-author, who reviewed our statistical methodology and informed the updated presentation of our data.

Reviewer Comment: Introduction, page 5, remove statements involving unpublished data related to C-26.

Author Response: Statements citing unpublished data have been edited to remove such references.

Reviewer Comment: The study rationale should be based off of published data and needs strengthened in the introduction. Please rework.

Author Response: The reference to unpublished data has been removed and the study rationale edited in the updated manuscript.

Reviewer Comment: The introduction hypothesis is not a hypothesis which is predictive and directional showing relationship between key variables. Please edit or restate as an objective.

Author Response: We thank the reviewer for the suggestion. The hypothesis was restated as an objective in the revised manuscript (see *Introduction*, p5, line 8, “Our objective was...”).

Reviewer Comment: The introduction's broad importance is cachexia, but the premise is delimited to the C-26 model. Please provide context for the C-26 model in relation to other preclinical models. Would your results with this model have implications for other models?

Author Response: We have updated the Introduction to place the C-26-model in the greater context of pre-clinical cachexia research (see *Introduction*, p4, “To date, several rodent models...”).

Reviewer Comment: Methods. Provide a reference and rationale for the AR-42 dosing paradigm and relevance of the doses in the dose response study.

Author Response: In the first section of the *Results*, the authors explain that the likely poor clinical tolerability of the original 50 mpk every other day anti-cachectic dose of AR-42 (J Natl Cancer Inst. 2015 Oct 12;107(12):d1v274.) drove our decision to evaluate lower doses (1-20 mpk, once daily). Our pharmacokinetic data suggested 10 mpk AR-42 (and the resulting 10.9 uM*h AUC) would be better tolerated in humans (Fig. 1A, Supplementary Fig. S1A) and we found this dose to be the lowest dose of AR-42 that provided anti-cachectic benefit (Fig. 1B, Supplementary Fig. S1B). Therefore, 10 mpk was carried forward to the remaining combination studies. We included additional dose-response data at the 1, 3 and 10 mpk levels to support our selection of 10 mpk for the combination treatment regimens.

In the updated manuscript, we have added some of this justification to the Materials and Methods section for clarity (see *Materials and Methods, AR-42 dose-response study* subsection, p28) and added results from the testing of the lower AR-42 doses to the data presented (see *Results*, p6, first paragraph, line 8, “Consequently, we evaluated the anti-cachectic effects of lower doses...”).

Reviewer Comment: Statistical methodology. Please supply more information in this section. Please clarify that ANOVA 2 way was performed (tumor and drug treatment). Please provide rationale for the multiple comparison tests used.

Author Response: In the reviewed version of the manuscript, the details of the specific statistical tests employed were primarily located in the figure legends and readers were pointed to the legends in the Materials and Methods section. As suggested, we have expanded our description of the statistical approaches used in the Materials and Methods section of the updated manuscript. As noted above, a one-way ANOVA was used to analyze our results. Our understanding is that a factorial design, in which all treatment groups are evaluated both with and without tumor burden, is best suited for a two-way ANOVA to evaluate potential interactions between drug treatments and tumor burden. In this work, only GTX-024 was evaluated in tumor-free groups and only in the male C-26 studies which limits the utility of this approach. Our one-way ANOVAs were followed with multiple comparisons tests (each group's mean compared to every other group's mean) with multiplicity adjusted p-values according to Tukey's methodology (Biometrics 48:1005-1013,1992) to control for multiple comparisons. For clarity, these details have been included in the Materials and Methods section, as well as in the figure legends in the updated manuscript.

Reviewer Comment: Page 9, please clarify that the dose "reversed", or did it block or prevent? To reverse means you have data that there was a significant drop that was then induced.

Author Response: The authors apologize for the confusing language. Our data only support the conclusion that treatment reduced changes, not reversed them. We have corrected this error in the updated manuscript (see *Results*, p6, first paragraph, line 11, “doses of 20 or 10 mg/kg AR-42 ameliorated...”)

Reviewer Comment: Please plot body weight change over time.

Author Response: Body weight changes over time are plotted in Figure 1D in the revised manuscript.

Reviewer Comment: Please add lean body mass measurements over time related to DEXA or similar methodology.

Author Response: As noted above in response to one of Reviewer #1's comments, the authors agree that longitudinal assessment of lean body mass would improve the manuscript. Regrettably, we do not have access to instrumentation that readily assesses lean mass in the models we present.

Reviewer Comment: Results, Figure 2. Please include data for AR-42 with no tumor.

Author Response: Data on tumor-free male CD2F1 mice receiving AR-42 have been previously published. (Tseng et al., J Natl Cancer Inst. 2015 Oct 12;107(12):djv274).

Reviewer Comment: Figure 4 A is a table. Please provide the N for the table data.

Author Response: Figure 4A has been removed and changed to **Table 1**. The data in this table and in the complete cytokine dataset presented in **Supplementary Table S1** are from the second C-26 study and the sera from all animals were analyzed (n=6 for tumor-free groups and n=7-10 for tumor-bearing groups). These details have been added to the table legends.

Reviewer Comment: Please clarify the relationship between figure 4D and Figure 4F, the graph appears more related to changes in total stat and the control is overloaded and not useful.

Author Response: The authors regret the confusion surrounding Figures 4D and 4F as they are unrelated. Fig. 4D is a western blot from gastrocnemius tissue and 4F is ELISA data from C-26 tumor tissue. In the revised manuscript, these figures are more clearly labeled in **Figure 6**.

Reviewer Comment: Were the changes in total STAT expected?

Author Response: We did not expect changes in total STAT.

2nd Editorial Decision

13 September 2019

Thank you for the submission of your revised manuscript to EMBO Molecular Medicine, and please accept my apologies for the delay in getting back to you, which is due to the fact that referee #1 did not respond to us regarding the re-evaluation of your manuscript.

In order not to delay the process further, we have now decided to make the decision based on referee #2's comments. As you will see from the report below, this referee is supportive of publication, but also mentions a number of issues that should be addressed in a minor revision of the present manuscript. To be clear, we would like you to discuss/address the concerns from referee #2, and if you do have data at hand, we would be happy for you to include it, however we will not ask you to provide any additional experiments at this stage.

I look forward to reading a new revised version of your manuscript as soon as possible.

***** Reviewer's comments *****

Referee #2 (Remarks for Author):

This reviewer had several concerns with the original submission that were centered on the manuscript's clarity, focus, and rigor.

While the reviewers have provided a thoughtful rebuttal that addresses many of the prior concerns, the predicted impact of the study could be improved by addition attention to several comments.

Overall in an era of renewed emphasis on transparency and rigor, the authors could more strongly present limitations of some endpoint measurements, and the lack of repeated measurement in the same animal over time.

Specific Comments

The revision has attempted to address prior concerns with statistical analysis.
 The presentation of body weight data has been improved.
 The data presentation has been reorganized to improve the clarity of the study
 The authors have improved the overall rigor of the study with the revised presentation. Detail has been added to the methods section.
 The authors have changed the working hypothesis to an objective
 Body weight over time is now presented.

Needs addressed

A more sophisticated measurement of lean body mass over time: Since the authors cannot perform this measurement, the authors need to include the limitations of the study for not being able to examine changes in lean body mass over the course of the study and how this effects the interpretation of endpoint measurements in the discussion.

The discussion has not been adequately streamlined and is still approximately 11 paragraphs, with the 2nd paragraph being 1.5 pages long. The discussion still spends considerable space repeating statements of results rather than discussing the impact on the field. The discussion should be further condensed, results removed, and main points emphasized, based on the current writing this

With the revision the authors have addressed the limitations of variability in circulating factors the study, however, the IL-6 data is not interpretable due to variability in tumor bearing mice without treatments there is no possible way an effect of treatment could be determined without a large increase in number of animals..
 could be accomplished in 8 normal paragraphs.

In figure 2 Please provide data on AR-42 with no tumor or remove as it does not have an appropriate control. The response saying that the control cannot be presented is not providing the control for this study. Please provide a value for the reader as dashed line with explanation or place in the text.

Remove data and results from discussion.
 Remove overuse of citing tables and figures in the discussion

Figures can be condensed
 Remove Figure 1C and place in text
 Remove Figure 1D, not interpretable
 Remove Figure 5C representative blot for negative data
 Figure 6B can be removed and the R and P place in the text.

2nd Revision - authors' response

26 November 2019

***** Reviewer's comments *****

Referee #2 (Remarks for Author):

This reviewer had several concerns with the original submission that were centered on the manuscript's clarity, focus, and rigor.

While the reviewers have provided a thoughtful rebuttal that addresses many of the prior concerns, the predicted impact of the study could be improved by addition attention to several comments.

Overall in an era of renewed emphasis on transparency and rigor, the authors could more strongly present limitations of some endpoint measurements, and the lack of repeated measurement in the same animal over time.

Specific Comments

The revision has attempted to address prior concerns with statistical analysis.
 The presentation of body weight data has been improved.
 The data presentation has been reorganized to improve the clarity of the study

The authors have improved the overall rigor of the study with the revised presentation. Detail has been added to the methods section.

The authors have changed the working hypothesis to an objective
Body weight over time is now presented.

1. Needs addressed A more sophisticated measurement of lean body mass over time: Since the authors cannot perform this measurement, the authors need to include the limitations of the study for not being able to examine changes in lean body mass over the course of the study and how this affects the interpretation of endpoint measurements in the discussion.

Author Response: Thank you for this comment. We have added a discussion of this additional limitation to our studies in the section of our discussion entitled “Impact of cachectic tumor burden on androgen signaling”.

2. The discussion has not been adequately streamlined and is still approximately 11 paragraphs, with the 2nd paragraph being 1.5 pages long. The discussion still spends considerable space repeating statements of results rather than discussing the impact on the field. The discussion should be further condensed, results removed, and main points emphasized, based on the current writing this could be accomplished in 8 normal paragraphs.

Author Response: We apologize that the discussion in our revised manuscript was insufficiently streamlined. To address this comment, we have further shortened and streamlined the discussion removing much of the repetition of results so that the discussion is more focused on the impact on the field.

3. With the revision the authors have addressed the limitations of variability in circulating factors the study, however, the IL-6 data is not interpretable due to variability in tumor bearing mice without treatments there is no possible way an effect of treatment could be determined without a large increase in number of animals.

Author Response: We thank the reviewer for his comment and appreciate the challenges associated with interpreting an apparent lack of changes when the control group is highly variable. We have updated the second paragraph of our results section entitled “Anti-cachectic efficacy of AR-42 is associated with STAT3 inhibition but not general immune suppression” to further highlight this notable limitation.

4. In figure 2 Please provide data on AR-42 with no tumor or remove as it does not have an appropriate control. The response saying that the control cannot be presented is not providing the control for this study. Please provide a value for the reader as dashed line with explanation or place in the text.

Author Response: We apologize for the confusion that our previous response to this inquiry may have caused. In our initial reply, we intended to point out that AR-42-mediated effects on relevant end points in a tumor-free animal were previously reported following multiple 50 mpk doses (J Natl Cancer Inst. 2015 Oct 12;107(12):d1v274, cited in the manuscript as Tseng et al., 2015.) No significant differences from controls were reported at this higher, maximally tolerated dose suggesting the lower dose of 10 mpk used throughout our report would not be expected to meaningfully impact any of these end points. We have added an expanded discussion of the effects of AR-42 in tumor-free mice from this published report to the first paragraph of the results section entitled “AR-42 administration demonstrates limited anti-cachectic effects at a reduced 10 mg/kg dose level”.

5. Remove data and results from discussion.

Author Response: We have removed nearly all the data and repetition of results from the discussion in our updated manuscript.

6. Remove overuse of citing tables and figures in the discussion

Author Response: We have removed the non-essential citing of figures and tables in the discussion.

7. Figures can be condensed
 A. Remove Figure 1C and place in text

Author Response: At the time of this manuscript's preparation there were more than 40 published descriptions of AR-42's effects in various models of disease. None of these papers provide a complete characterization of this agent's effects on various HDAC isoforms. As this agent remains under active clinical development, we believe sharing this data will be of broad interest to the readership of *EMBO Molecular Medicine* and especially to those with an interest in additional indications for this novel HDAC inhibitor. To address the wishes of the reviewer and still provide access to this data, we have summarized these findings in the text (p6) and also provide the full results in the updated appendix (Appendix Figure S1C).

- B. Remove Figure 1D, not interpretable

Author Response: The authors have removed this figure in our updated manuscript but want to be certain this is acceptable to the reviewer as this is the only figure where changes in bodyweight are presented over time. This specific representation of the data was noted previously by the reviewer as an important improvement over previous drafts.

- C. Remove Figure 5C representative blot for negative data

Author Response: We agree that this blot serves primarily as negative data but it also supports androgen receptor stabilization in the presence of agonist which shows evidence of pharmacologically relevant amounts of GTx-024 in gastrocnemius tissue. This is discussed in the second paragraph of the results section entitled "Effects of tumor burden and GTx-024/AR-42 treatment on the expression of AR and atrophy-related genes in skeletal muscle". To address the reviewer's request, we will remove this figure from the main manuscript. As we feel this data is still sufficiently important to be presented, we have moved this blot to the appendix as Appendix Figure S5.

- D. Figure 6B can be removed and the R and P place in the text.

Author Response: Thank you for this comment. We have removed the figure and simply placed the R and P values in the text as requested (p14).

Corresponding Author Name: Christopher C. Coss

Manuscript Number: EMM-2018-09910-V2